# DiffSketcher: Text Guided Vector Sketch Synthesis through Latent Diffusion Models

**Ximing Xing**
Beihang University
ximingxing@buaa.edu.cn

**Chuang Wang**
Beihang University
chuangwang@buaa.edu.cn

**Haitao Zhou**
Beihang University
zhouhaitao@buaa.edu.cn

**Jing Zhang**
Beihang University
zhang_jing@buaa.edu.cn

**Qian Yu**[*]
Beihang University
qianyu@buaa.edu.cn

**Dong Xu**
The University of Hong Kong
dongxu@cs.hku.hk

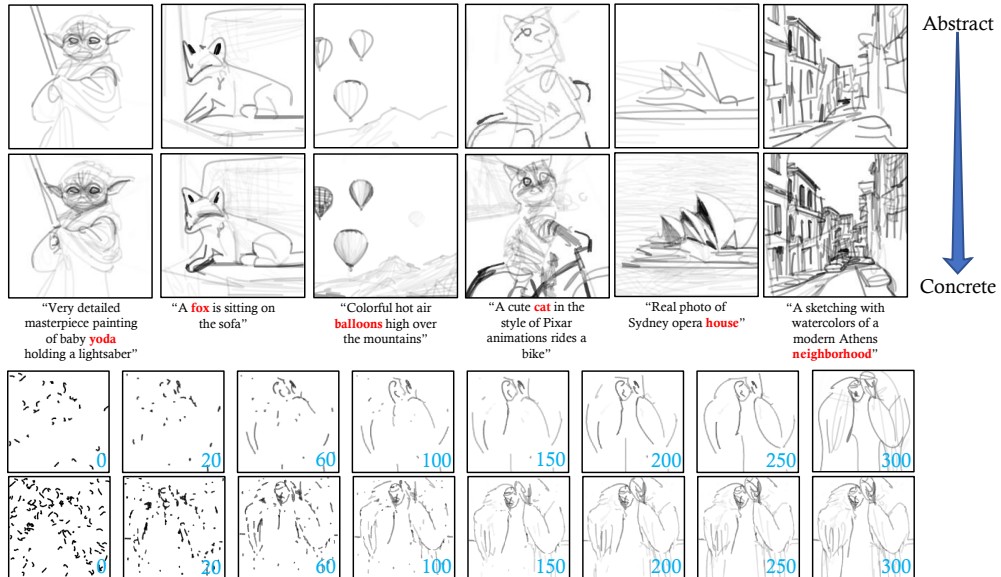

Abstract

Concrete

"Very detailed masterpiece painting of baby **yoda** holding a lightsaber"

"A **fox** is sitting on the sofa"

"Colorful hot air **balloons** high over the mountains"

"A cute **cat** in the style of Pixar animations rides a bike"

"Real photo of Sydney opera **house**"

"A sketching with watercolors of a modern Athens **neighborhood**"

"**Macaw** full color, ultra detailed, realistic, insanely beautiful"; blue number indicates the number of iterations

Figure 1: **Top**: Visualizations of the vector sketches generated by our proposed method, *DiffSketcher*. **Bottom**: Visualizations of the drawing process. For each example, we show two sketches with a different number of strokes.

## Abstract

Even though trained mainly on images, we discover that pretrained diffusion models show impressive power in guiding sketch synthesis. In this paper, we present DiffSketcher, an innovative algorithm that creates *vectorized* free-hand sketches using natural language input. DiffSketcher is developed based on a pre-trained text-to-image diffusion model. It performs the task by directly optimizing a set of Bézier curves with an extended version of the score distillation sampling (SDS) loss, which allows us to use a raster-level diffusion model as a prior for optimizing a parametric vectorized sketch generator. Furthermore, we explore attention maps embedded in the diffusion model for effective stroke initialization to speed up

---

[*]Corresponding author

37th Conference on Neural Information Processing Systems (NeurIPS 2023).

the generation process. The generated sketches demonstrate multiple levels of abstraction while maintaining recognizability, underlying structure, and essential visual details of the subject drawn. Our experiments show that DiffSketcher achieves greater quality than prior work. The code and demo of DiffSketcher can be found at https://ximinng.github.io/DiffSketcher-project/.

# 1   Introduction

Minimal representations, such as sketches and natural language, are powerful tools for effectively conveying ideas by emphasizing the subject's essence. As the carriers of abstract concepts, natural language conveys abstract semantic understanding, whereas sketches embody the human visual abstract presentation. A sketch can provide more visual details than natural language, making the concept more concrete. When a designer discusses the design plan with a client, the designer may sketch a prototype based on the client's description to ensure a full understanding of his/her requirements. If the process of text-to-sketch could be learned automatically, it would significantly lessen associated labor costs.

Unfortunately, the task of text-to-sketch generation remains unexplored. Some studies conducted on generating image-conditioned sketches [3, 18, 45, 44]. One such method is Info-drawing [3], which treats a sketch as a photo and uses a generative adversarial network (GAN) to generate a raster sketch based on an input image. It also introduces a style loss to ensure the generated sketches have a similar style to the reference sketch used in training. CLIPasso [45] presents a novel pipeline to produce vector sketches through a differentiable rasterizer optimized by a semantic loss. CLIPasso can only handle the images with a single object, so its follow-up work CLIPascene [44] extends to scene-level images. However, all of these methods depend on input images and do not facilitate the generation of sketches directly from text inputs. Moreover, despite their ability to generate realistic sketches from photos, these methods have a limitation in that they cannot generate *new content*.

Recent breakthroughs in text-to-image generation have been driven by diffusion models [23, 28, 30, 31] trained on billions of image-text pairs [34]. These models, conditioned on text, now support high-fidelity, diverse, and controllable image synthesis. Nonetheless, the present text-to-image diffusion models cannot produce highly abstract and vectorized free-hand sketches (as shown in Fig. 2). We draw inspiration from the highly effective text-to-image diffusion models and image-conditioned sketch generators to build a bridge between two fundamental forms of human expression, namely text and free-hand sketches, resulting in the development of our proposed text-to-sketch generator.

In this work, we present DiffSketcher, an algorithm that can synthesize novel and high-quality free-hand vector sketches based on natural language text input. DiffSketcher does not require any text-to-sketch training pairs or large datasets of sketches. Instead, with the guidance of a pretrained diffusion model [30], we define a sketch as a set of Bézier curves and optimize the curves' parameters through a differentiable rasterizer [16]. The key idea behind DiffSketcher is to transfer the prior knowledge from text-to-image generation model into the differentiable rasterizer. This allows the optimization of text-guided sketches with semantic consistency, ensuring that the final output sketches are coherent and aligned with their corresponding text inputs. However, it is a non-trivial task to fully leverage the pre-trained diffusion model to efficiently generate high-quality free-hand sketches for both simple objects and complex scenes.

To this end, we propose three strategies to improve the generation quality and efficiency: (1) Present an extended Score Distillation Sampling (SDS) Loss to guide the optimization of curve parameters. Previous works optimize the parameters of the vector sketch with CLIP loss. We found that an extended version of SDS loss can give more diverse sketch synthesis results, and it can be combined with CLIP loss or LPIPS loss, providing an additional control term. (2) Explore attention maps embedded in the diffusion model for effective stroke initialization. The synthesis process can be very time-consuming if the starting point of each stroke is randomly initialized. Therefore, we explore different initialization strategies and present to initialize the curve control points with a fused product of cross-attention maps and self-attention maps in the U-Net [5] of the diffusion model, which significantly improves the efficiency compared to the random initialization. (3) Introduce opacity property during the optimization of Bézier curves to better mimic the style of human sketches, achieving the effect of heavy and light brushstrokes.

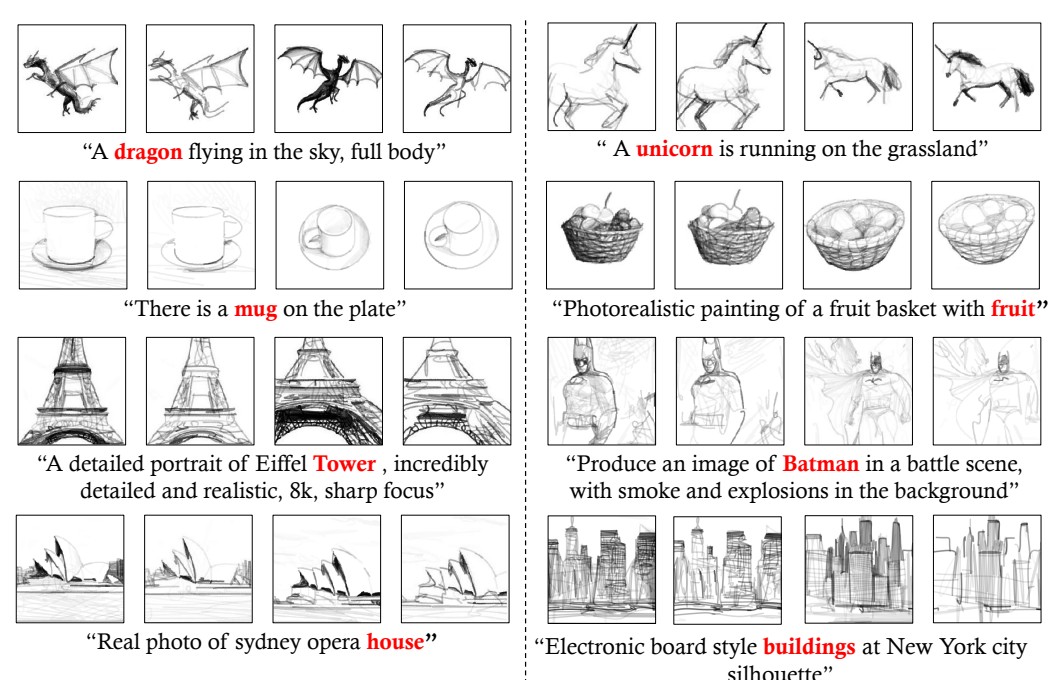

"A **dragon** flying in the sky, full body"

" A **unicorn** is running on the grassland"

"There is a **mug** on the plate"

"Photorealistic painting of a fruit basket with **fruit**"

"A detailed portrait of Eiffel **Tower** , incredibly detailed and realistic, 8k, sharp focus"

"Produce an image of **Batman** in a battle scene, with smoke and explosions in the background"

"Real photo of sydney opera **house**"

"Electronic board style **buildings** at New York city silhouette"

Figure 2: Various free-hand sketches synthesized by DiffSketcher and the corresponding description prompts. DiffSketcher obtains prior information from LDM [30] composite images through score distillation [24] and achieves the same heavy and light drawing styles as human sketches by performing gradient descent on a set of Bézier curves with the opacity property. Our proposed DiffSketcher allows for varying levels of abstraction while matching its corresponding textual semantics. In each example, given the same text prompt and two different random seeds, two sketches with a different number of strokes are generated. The red words represent the cross-attention index used to initialize the control points (details about cross-attention are covered in Section 4.2).

In summary our contributions are three-fold: (1) We propose a text-to-sketch diffusion model dubbed DiffSketcher. To the best of our knowledge, it is the first diffusion model to generate diverse and high-quality vector sketches with different levels of abstraction at the object and scene levels. (2) We present three strategies to improve the generation quality and efficiency, including an extended SDS loss and an attention-enhanced stroke initialization strategy. The opacity property of strokes is considered during synthesis to improve the visual effect further. (3) We conduct extensive experiments, and the experimental results show that our model outperforms other models in sketch quality and diversity. Thorough insights are provided for future research.

## 2   Related Work

**Sketch Synthesis.**   Free-hand drawings convey abstract concepts through human visual perception with minimal abstraction. Unlike purely edge-map extraction methods [2], free-hand sketching aims to present sketches that are abstract in terms of structure [3] and semantic interpretation[45]. Therefore, computational sketching methods that aim to mimic human drawing consider a wide range of sketch representations, ranging from those grounded in the edge map of the input image[46, 17, 15, 42, 18, 4] to those that are more abstract [9, 8, 1, 7, 26, 45], which are normally in vector format. Among the works synthesizing vector sketches, CLIPasso [45] and CLIPascene [44] are conditioned on an input image, while the rest are unconditional. Until now, no prior work has explored synthesizing a sketch based on text.

**Vector Graphics.**   Our work builds upon the differentiable renderer for vector graphics introduced by Li et al [16]. While image generation methods that operate over vector images traditionally require a vector-based dataset, recent work has shown how the differentiable renderer can be used to bypass this limitation [36, 14, 41, 29, 19]. Furthermore, recent advances in visual text embedding contrastive language-image pre-training (CLIP)[27] have enabled a number of successful methods for synthesizing sketches, such as CLIPDraw [7], StyleCLIPDraw [32], CLIP-CLOP [21], and CliPascene [44]. A very recent work VectorFusion [13] combine differentiable renderer with diffusion

model for vector graphics generation, *e.g.*, iconography and pixel art. Our proposed algorithm, DiffSketcher, shares a similar idea with VectorFusion, but our focus is generating object- and scene-level sketches from a natural language prompt.

**Diffusion Models.** Denoising diffusion probabilistic models (DDPMs) [37, 39, 11, 40], particularly those conditioned on text, have shown promising results in text-to-image synthesis. For example, Classifier-Free Guidance (CFG) [12] has improved sample quality and has been widely used in large-scale diffusion model frameworks, including GLIDE [23], Stable Diffusion [30], DALL·E 2 [28], and Imagen [31]. However, the majority of images available in web-scale datasets are rasterized, and this work follows the framework of *synthesis through optimization*, in which images are generated through evaluation-time optimization against a given metric. Our proposed algorithm, *DiffSketcher*, uses a pre-trained text-to-image diffusion model to synthesize free-hand sketches from natural language input. This is achieved by transferring image synthesis prior information into a differentiable renderer.xc

## 3 Preliminaries

### 3.1 Diffusion Models

In this section, we provide a concise overview of diffusion models, which are a class of generative models that utilize latent variables to gradually transform a sample from a noise distribution to a target data distribution [37, 11]. Diffusion models consist of two components: a forward process $q$ and a reverse process or generative model $p$. The forward process, which is typically modeled as a Gaussian distribution, gradually removes structure from the input data $\boldsymbol{x}$ by adding noise over time. The reverse process, on the other hand, adds structure to the noise starting from a latent variable $\boldsymbol{z}_t$. Specifically, the generative model is trained to slowly add structure starting from random noise $p(\boldsymbol{z}_T) = \mathcal{N}(\boldsymbol{0}, \boldsymbol{I})$ with transitions $p_\phi(\boldsymbol{z}_{t-1}|\boldsymbol{z}_t)$. $q(\boldsymbol{z}_t|\boldsymbol{x}) = \mathcal{N}(\alpha_t\boldsymbol{x}, \sigma_t^2\boldsymbol{I})$. Training the diffusion model with a (weighted) evidence lower bound (ELBO) simplifies to a weighted denoising score matching objective for parameters $\phi$ [11]

$$\mathcal{L}_{\text{Diff}}(\phi, \boldsymbol{x}) = \mathbb{E}_{t\sim\mathcal{U}(0,1),\epsilon\sim\mathcal{N}(\boldsymbol{0},\boldsymbol{I})}\left[w(t)\left\|\epsilon_\phi\left(\alpha_t\boldsymbol{x} + \sigma_t\epsilon; t\right) - \epsilon\right\|_2^2\right] \tag{1}$$

where $w(t)$ is a weighting function that depends on the timestep $t$. Our work builds on text-to-image latent diffusion model (LDM) that learn $\epsilon_\phi(\boldsymbol{z}_t; t, y)$ conditioned on text embeddings $y$ [30]. LDM uses classifier-free guidance(CFG) [12], which jointly learns an unconditional model to enable higher quality generation via a guidance scale parameter $\omega : \hat{\epsilon}_\phi(\boldsymbol{z}_t; y, t) = (1 + w)\epsilon_\phi(\boldsymbol{z}_t; y, t) - w\epsilon_\phi(\boldsymbol{z}_t; t)$ ($\hat{\epsilon}_\phi$ denotes the guided version of the noise prediction). CFG alters the score function to prefer regions where the ratio of the conditional density to the unconditional density is large. In practice, setting $w > 0$ improves sample fidelity at the cost of diversity.

### 3.2 Score Distillation Sampling

Many 3D generative approaches use a frozen image-text joint embedding model (*e.g.* CLIP) and an optimization-based approach to train a Neural Radiance Fields (NeRF) [20]. Such models can be specified as a differentiable image parameterization (DIP) [22], where a differentiable generator $g$ transforms parameters $\theta$ to create an image $\boldsymbol{x} = g(\theta)$. In DreamFusion [24], $\theta$ be parameters of a 3D volume and $g$ is a volumetric renderer. To learn these parameters, DreamFusion proposed *score distillation sampling* (SDS) loss that can be applied to Imagen [31]:

$$\nabla_\theta\mathcal{L}_{\text{SDS}}(\phi, \boldsymbol{x} = g(\theta)) \triangleq \mathbb{E}_{t,\epsilon}\left[\omega(t)(\hat{\epsilon}_\phi(\boldsymbol{z}; y, t) - \epsilon)\frac{\partial\boldsymbol{x}}{\partial\theta}\right] \tag{2}$$

where the constant $\alpha_t\boldsymbol{I} = \partial\boldsymbol{z}_t/\partial\boldsymbol{x}$ is absorbed into $w(t)$, and the classifier-free-guided $\hat{\epsilon}_\phi$ is used. In practice, SDS gives access to loss gradients, not a scalar loss. Their proposed SDS loss provides a way to assess the similarity between an image and a caption:

$$\nabla_\theta\mathcal{L}_{\text{SDS}}(\phi, \boldsymbol{x} = g(\theta)) = \nabla_\theta\mathbb{E}_t\left[\sigma_t/\alpha_t w(t)\text{KL}\left(q\left(\boldsymbol{z}_t|g(\theta); y, t\right)|p_\phi\left(\boldsymbol{z}_t; y, t\right)\right)\right] \tag{3}$$

where $p_\phi$ is the distribution learned by the frozen, pretrained Imagen model. $q$ is a unimodal Gaussian distribution centered at a learned mean image $g(\theta)$. DreamFusion [24] proposed an approach to use a pretrained pixel-space text-to-image diffusion model (Imagen [31]) as a loss function. However,

diffusion models trained on pixels have traditionally been used to sample only pixels. We want to create what look like good sketches that match the text prompts when rendered from a set of vector strokes. Such models can be specified as a differentiable image parameterization, where a differentiable rasterizer $\mathcal{R}$ transforms parameters $\theta$ to create a sketch $\mathcal{S} = \mathcal{R}(\theta)$.

Inspired by DreamFusion [24], we extend *score distillation sampling* (SDS) loss to use a pretrained latent diffusion model [30] as a prior for optimizing curve parameters. Intuitively, score distillation converts diffusion sampling into an optimization problem that allows the raster image to be represented by a differentiable rasterizer.

### 3.3 Differentiable Rasterizer

Li *et al.* [16] introduce a differentiable rasterizer $\mathcal{R}$ that bridges the vector graphics and raster image domains. A raster image is a 2D grid sampling over the space of the vector graphics scene $f(x, y; \Theta)$, where $\Theta$ contains the curve parameters, *e.g.,* coordinates of Bézier control points, opacity, line thickness. Given a 2D location $(x, y) \in \mathbb{R}^2$, they first find all the filled curves and strokes that overlap with the location. Then sort them with a user-specified order and compute the color using alpha blending [25]. It enables the gradients of the rasterized curves (image) to be backpropagated to the curve parameters.

## 4  Methodology

In this section, we present our method for generating sketches using pre-trained text-to-image diffusion models. Let $\mathcal{S}$ be a sketch that was rendered by a differentiable rendering [16] $\mathcal{R}$ using the text prompt $\mathcal{P}$. Our goal is to optimize a set of parametric strokes to automatically generate a vector sketch that matches the description of the text prompt.

Our pipeline is illustrated in Figure 3. Given a text prompt $y$ of the desired subject and a set of control points for the strokes, we synthesize the corresponding sketch $\mathcal{S}$ while matching the semantic attributes of the text prompt. To initialize the control points of the strokes, we extract and fuse the attention map of the U-Net [5] used by the latent diffusion model [30]. Our key observation is that the structure and appearance of the generated image depend on the interaction between the pixels and the text embedding through the diffusion process [10]. We provide more details on strokes initialization in Section 4.2.

As shown in Figure 4, in each step of the optimization, we feed the stroke parameters to a differentiable rasterizer $\mathcal{R}$ to produce the raster sketch. We optimize over parameters $\theta$ such that $\mathcal{S} = \mathcal{R}(\theta)$ is close to a sample from the frozen latent diffusion model [30]. To perform this optimization, we propose a variation of the SDS loss function that enhances the texture of hand-sketched strokes and improves the resilience of drawing styles. This results in the raster sketch always being coherent with the text prompt. The resulting sketch, as well as the sample from the frozen latent diffusion model, then defines a joint semantic and perceptual loss. We back-

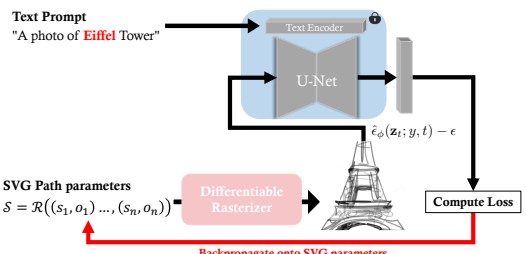

Figure 3: The overview of the pipeline. DiffSketcher accepts a set of control points (the locations of the strokes) and text prompts as input to generate a hand-drawn sketch.

propagate the loss through the differentiable rasterizer $\mathcal{R}$ and update the control points and opacity of strokes directly at each step until convergence of the loss function.

### 4.1  Synthesis Through Optimization

We define a sketch as a set of $n$ strokes $\{s_1, \ldots, s_n\}$ with the opacity attribute placed on a white background. To represent each stroke, we use a two-dimensional Bézier curve with four control points $\mathbf{s}_i = \{p_i^j\}_{j=1}^4 = \{(x_i, y_i)^j\}_{j=1}^4$ (Notice that here, $x$ and $y$ represent the coordinates in the canvas) and one opacity attribute $o_i$. We incorporate the opacity of the strokes into the optimization process and use DiffSketcher semantics understanding to achieve a human-like *heavy and light sketch style*. The parameters of the strokes are fed to a differentiable rasterizer $\mathcal{R}$, which forms the raster

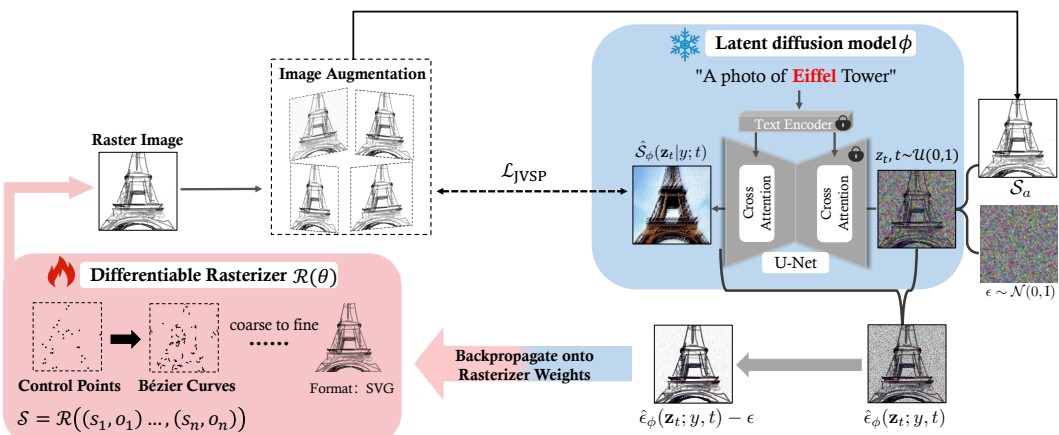

Figure 4: Optimization overview. To synthesize a sketch that matches the given text prompt, we optimize the parameters of the differentiable rasterizer $\mathcal{R}$ that produces the raster sketch $\mathcal{S}$, such that the resulting sketch is close to a sample from the frozen latent diffusion model (the blue part of the picture). Since the diffusion model directly predicts the update direction, we do not need to backpropagate through the diffusion model; the model simply acts like an efficient, frozen critic that predicts image-space edits.

sketch $\mathcal{S} = \mathcal{R}((s_1, o_1) \ldots, (s_n, o_n)) = \mathcal{R}((\{p_1^j\}_{j=1}^4, o_1), \ldots, (\{p_n^j\}_{j=1}^4, o_n))$. For simplicity, we define the parameter in $\mathcal{R}$ as $\theta$.

### 4.1.1 Vanilla Version: Fidelity to Generated Image

We start with a two-stage pipeline: First, we sample an image from the latent diffusion model [30] using a text prompt. Next, we optimize the control points to obtain a sketch that is consistent with the text prompt. To preserve the fidelity of the generated sample, we incorporate a **J**oint **V**isual **S**emantic and **P**erceptual (JVSP) loss to optimize the similarity of the synthesized sketches and the instance sampled from the frozen latent diffusion model [30]. We leverage the VAE [6] decoder $\mathcal{D}$ to get the RGB pixel representation of $\hat{\mathcal{S}}_\phi(\mathbf{z}_t|y;t)$. Then, we jointly use the LPIPS [47] and CLIP visual encoders [27, 45] as depth structural similarity and visual semantic similarity metrics, respectively. Specifically, we use the following loss function:

$$\mathcal{L}_{\text{JVSP}} = \mathcal{L}_{\text{LPIPS}}(\mathcal{D}(\hat{\mathcal{S}}_\phi(\mathbf{z}_t|y;t)), \mathcal{R}(\theta)) + \sum_l \left\| \text{CLIP}_l(\mathcal{D}(\hat{\mathcal{S}}_\phi(\mathbf{z}_t|y;t))) - \text{CLIP}_l(\mathcal{R}(\theta)) \right\| \quad (4)$$

The $\mathcal{L}_{\text{JVSP}}$ loss function encourages the synthesized sketch to match the underlying semantics and perceptual details of the image sampled from LDM, leading to more realistic and visually appealing results. While vectorizing a rasterized diffusion sample is lossy, an input **A**ugmentation version of the **SDS** (ASDS) loss can either finetune the results or optimize random control points from scratch to sample a sketch that is consistent with the text prompt. In the Sec 4.1.2, we introduce the ASDS loss to match the text prompt.

### 4.1.2 Augmentation SDS Loss: Fidelity to Text Prompt

To synthesize a vector sketch that matches a given text prompt, we directly optimize the parameters $\theta$ of the differentiable rasterizer $\mathcal{R}$ that produces the raster sketch $\mathcal{S}$. We propose an input augmentation version of the SDS loss function to perform this optimization, which encourages plausible images to have low loss and implausible images to have a high loss. Given a raster sketch $\tilde{\mathcal{S}} \in \mathbb{R}^{H \times W \times 3}$, we combine *RandomPerspective*, *RandomResizedCrop* and *RandomAdjustSharpness* to get a data augmentation version of $\tilde{\mathcal{S}}_a \in \mathbb{R}^{512 \times 512 \times 3}$. These transformations preserve artistic styles, enhance style diversity, emphasize crucial features, and improve the model's adaptability to various artistic renditions and size variations commonly found in black-and-white hand-drawn sketches. Then, the LDM uses a VAE encoder [6] to encode $\tilde{\mathcal{S}}_a$ into a latent representation $\mathbf{z} = \mathcal{E}(\tilde{\mathcal{S}}_a)$, where $\mathbf{z} \in \mathbb{R}^{(H/f) \times (W/f) \times 4}$ and $f$ is the encoder downsample factor. In summary, we use the following

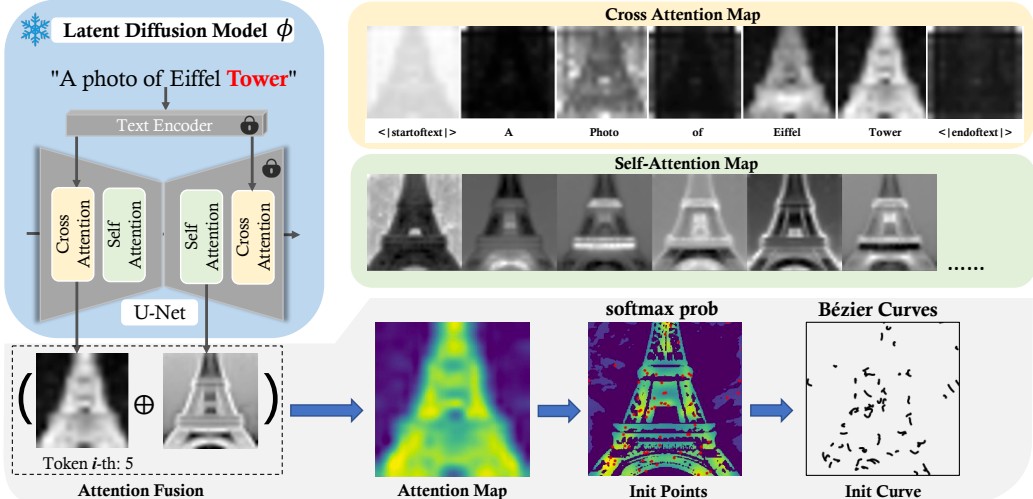

Figure 5: Strokes Initialization. The blue part of the figure represents the UNet in the LDM, which has two types of attention mechanisms: self-attention and cross-attention. The yellow and green parts respectively depict the visualization results of the cross-attention and self-attention. The gray part shows how the initial strokes are generated using a fused attention map. The dashed box represents the attention fusion, which is composed of the mean of the self-attention map and the cross-attention map corresponding to the 5-th text prompt token ("Tower"). We start at 1-th token, because 0-th token is taken up by the CLIP starting token.

ASDS loss function:

$$\nabla_\theta \mathcal{L}_{\mathrm{ASDS}}(\phi, \mathcal{S} = \mathcal{R}(\theta)) \triangleq \mathbb{E}_{t,\epsilon,a} \left[ w(t)(\hat{\epsilon}_\phi(\boldsymbol{z}_t; y, t) - \epsilon) \frac{\partial \boldsymbol{z}}{\partial \tilde{\mathcal{S}}_a} \frac{\partial \tilde{\mathcal{S}}_a}{\partial \theta} \right] \tag{5}$$

Where the weighting function $w(t)$ is a hyper-parameter. And we sample $t \sim \mathcal{U}(0.05, 0.95)$, avoiding very high and low noise levels due to numerical instabilities. Like DreamFusion [24], we set $\omega = 100$ for classifier-free guidance and higher guidance weights give improved sample quality. Intuitively, this loss perturbs $\tilde{\mathcal{S}}_a$ with a random amount of noise corresponding to the timestep $t$, and estimates an update direction that follows the score function of the diffusion model to move to a higher density region. The ASDS loss function encourages the synthesized sketch to match the given text prompt, while also preserving the style and structure of the original sketch. At each iteration, we backpropagate the loss through the differentiable rasterizer and update the control points and opacity of strokes directly until convergence of the loss function.

**Loss objectives of our final model.** To further enhance the quality of the synthesized sketches, we incorporate the JVSP and ASDS losses. Specifically, we first obtain the initial results with the JVSP loss, then we fine-tune the differentiable rasterizer together with the ASDS loss. We found such design can achieve the best performance. The ASDS loss predicts the gradient update direction directly, thus its loss value does not require weight balancing. In the JVSP loss, we set the weight of the LPIPS item to 0.2, and the weight of the CLIP visual item to 1. To compute the L2 distance between intermediate level activations of CLIP, we follow the CLIPasso [44] method and use layers 3 and 4 of the ResNet101 CLIP model. As shown in Figure 8, we compare the performance of the vanilla version with that of the version using only the ASDS loss, and the proposed final version. Our experiments show that the final version improves both the generation quality and efficiency.

## 4.2 Joint Attention-based Stroke Initialization

The highly non-convex nature of the ASDS loss function makes the optimization process susceptible to initialization, especially in multi-instance scenarios where strokes must be carefully placed to emphasize the overall semantics conveyed by free-hand sketching. We improve the convergence towards semantic depictions by initializing the curve control points based on the attention map of the text prompt conditional latent diffusion model. The UNet in the LDM has two types of attention mechanisms (the blue part), self-attention and cross-attention, as shown in Figure 5 where the yellow part represents the cross-attention visualization result, and the green part represents the self-attention visualization result. The structure and appearance of the image generated by the LDM [30] depend on

the interaction between the pixels to the text embedding through the diffusion process [10], which is manifested in the cross-attention layer [43]. The visualization results indicate that the cross-attention layers control the relationship between the spatial layout of the image and each word in the prompt, while the self-attention layer affects the spatial layout and geometric structure of the generated image. With this observation, we linearly combine the probability distributions of the two attention maps to initialize the curve control points. Specifically, we pick a map in the cross-attention maps based on a text token and combine it with the averaged self-attention map. This process is formalized as $\text{FinalAttn} = \lambda * \text{CrossAttn}_i + (1 - \lambda) * \text{Mean}(\text{SelfAttn})$, where $\lambda$ is the control coefficient and $i$ indicates the $i$-th token in the text prompt. Finally, we normalize the fused attention map using softmax and use it as a distribution map to sample $n$ positions for the first control point $p_i^1$ of each Bézier curve. The other three control points $(p_i^2, p_i^3, p_i^4)$ are sampled within a small radius (0.05 of image size) around $p_i^1$ to define the initial set of Bézier curves $\{\{p_i^j\}_{j=1}^4\}_{i=1}^n$. Empirical results show that our attentional fusion-based initialization contributes significantly to the quality and rendering speed of the final sketch in comparison to the random initialization.

# 5  Results

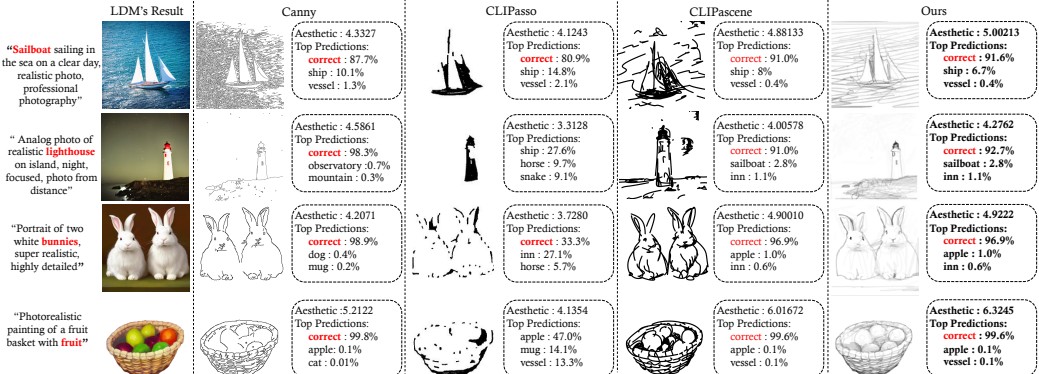

Figure 6: Comparison with existing methods, including edge extraction [2] and vector sketching [45, 44].

## 5.1  Qualitative Evaluation

As shown in Figure 2, we demonstrate that our approach offers the ability to produce object-level and scene-level sketches based on a textual prompt, with the flexibility to manipulate the level of abstraction through the number of strokes. It is effective in generating accurate sketches regardless of prompt complexity, including simple or non-existent objects, iconic constructions, and detailed scenes. The utilization of the robust prior of stable diffusion allows for a favorable initialization, promoting the production of high-quality sketches with significantly fewer iterations. Figure 6 qualitatively compared our DiffSketcher with those of CLIP-based [45] and edge extraction-based methods [2]. The Canny edge extraction algorithm extracts an excessive number of edges and produces untidy sketches, as observed in the first example in Fig. 6. CLIPasso uses visual distance metrics to guide gradient-based optimization. On the task of drawing scene-level sketches, CLIPasso can only draw part of the foreground, and the background part is missing.

We also compare our work with the recent work VectorFusion [13]. The results are shown in Fig. 7. This work is highly relevant to ours as it also explores the potential of diffusion model in generating vector graphics. However, there are notable differences between our appraoch and VectorFusion in terms of task setting, model design, and performance. Firstly, our method, DiffSketcher, primarily focuses on generating vector sketches based on text input. In contrast, VectorFusion aims to generate a broader range of vector graphics, including iconography and pixel art. It is important to highlight that our method has the capability to easily extend its functionality to generate other types of vector graphics *without* changing primitives (as shown in Fig. 10 of Supp. B). Secondly, DiffSketcher follows a distinct pipeline compared to VectorFusion. VectorFusion employs different model variants for different types of vector graphics. For generating vector sketches, our approach differs from VectorFusion mainly in two aspects: (1) **Initialization**: VectorFusion randomly initializes the

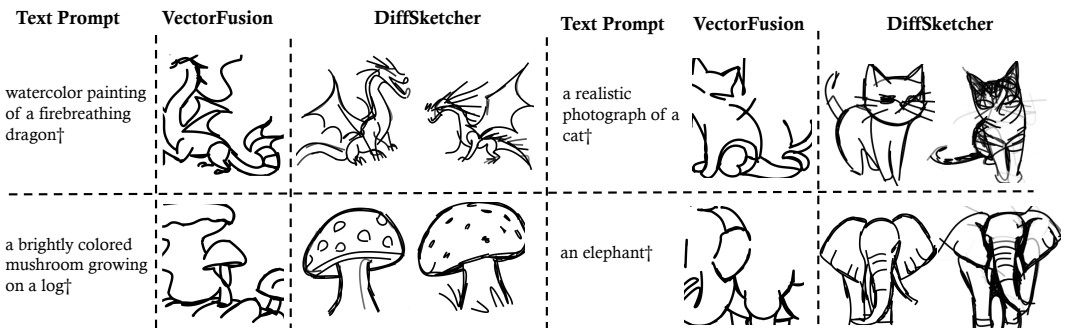

| Text Prompt | VectorFusion | DiffSketcher | Text Prompt | VectorFusion | DiffSketcher |

Figure 7: Qualitative comparison with VectorFusion(VF) [13]. VF's results are based on its original paper and project website, with a text prompt suffix of "minimal 2D line drawing trending on ArtStation". Note that our results were optimized from scratch using ASDS *without* specifically designed text prompt suffix.

rasterizer (i.e., the location of the control points), while our approach utilizes the attention layer of LDM, guided by text prompts, for initialization. This significantly improve both efficiency and synthesis quality. (2) **Optimization**: While VectorFusion solely optimizes the position of control points, our method also optimize the opacity of stroke. This contributes to enhance the visual quality of the synthesized sketches. As shown in Fig. 7, our approach significantly outperforms VectorFusion in generating vector sketches, and achieves comparable performance in generating other types of vector graphics (see Fig. 10 of Supp. B). It is worth noting that our model variant shown in Fig. 7 utilizes only ASDS loss for optimization and initializes the strokes randomly. This was done to ensure a fair comparison with VectorFusion. Our full model is expected to deliver better performance.

## 5.2 Quantitative Evaluation

Evaluating text-to-sketch synthesis is challenging due to the absence of ground truth sketches. Therefore, we focus on three indicators: consistency between the generated sketch and text prompt, the aesthetic quality of the sketch, and the recognition accuracy of the sketch. To measure the consistency between the generated sketch and input text, we calculate the mean of cosine similarity of CLIP embeddings for the generated sketches and the text captions used to generate them. Our method achieves a cosine similarity of 0.3494, which is higher than the 0.328 achieved by Canny algorithm and the 0.3075 achieved by CLIPasso. Aesthetic appeal is subjective, and being visually appealing is a personal experience and preference. However, when describing the visual appeal of a hand-drawn sketch, various factors can be considered, such as line quality, texture, and style. As an evaluation of our proposed method, we use CLIP-based aesthetic indicators [33] to calculate the aesthetic value for samples of multiple categories. Figure 6 compares aesthetic values achieved by various methods. Our method achieves a mean value of 4.8206 for the aesthetic value on a large number of examples, which is higher than the 4.3682 achieved by Canny and the 4.0821 achieved by CLIPasso [45].

## 5.3 Ablation Study

We conducted a series of experiments to demonstrate the effectiveness of the proposed initialization strategy and the effects of ASDS loss and JVSP loss, respectively. The top row of Fig. 8 compares two initialization strategies used by DiffSketcher and CLIPasso [45], namely LDM Attention and CAM [35] Saliency map, respectively. The text-driven diffusion model produces more precise attention map than the saliency map obtained by the CLIP-based method due to LDM's superior generation ability. The cross-attention feature of LDM can efficiently activate relevant regions based on token areas, while the self-attention layer can effectively differentiate foreground from background down to the pixel level. Through combining both mechanisms, the initialization area becomes more precise. As a result, superior quality sketches are produced with a reduced number of optimization steps, as illustrated in the second and third rows of Fig. 8. The proposed initialization strategy improves sampling quality and efficiency, which is critical for non-convex objective function optimization.

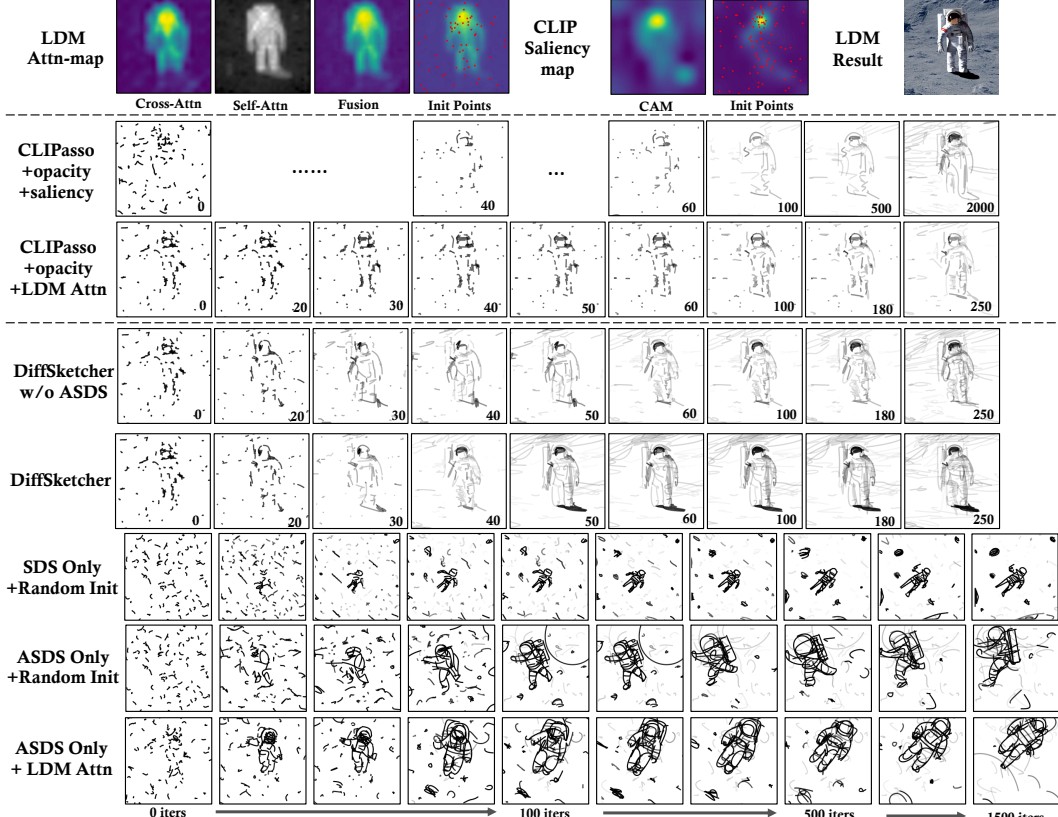

Figure 8: Qualitative results of ablation study. **Top**: two initialization strategies used by DiffSketcher (*i.e.*, LDM Attention) and CLIPasso [45] (*i.e.*, CAM [35] Saliency map). A result of sampling from LDM . The **2nd** and **3rd** rows: Comparison of the two initialization strategies on the convergence speed. The **4th** and **5th** rows: The effect of JVSP (Section 4.1.1) and ASDS loss (Section 4.1.2). The **5th** row shows the loss of SDS without data augmentation. The **6th** and **7th** rows: ASDS loss leads to more diverse results when strokes are randomly initialized. Text prompt used in this example: "Astronaut on Asteroid, galaxy background".

In DiffSketcher, our JVSP loss consists of a CLIP loss and a LPIPS loss, and it is important to note that ASDS and JVSP do not conflict with each other. As illustrated in Fig. 8, the 4th and 8th rows highlight the effects of JVSP loss and ASDS loss, respectively. When only JVSP loss is used (4th row), the generated sample closely approximates the result of LDM. On the other hand, when only ASDS loss is employed (8th row), the generated sample aligns with the text prompt semantically, but does not follow the attention map of the LDM. Additionally, using ASDS loss results in more diverse sampling outcomes during the synthesis process. For instance, the location of the astronaut's head may change throughout the drawing process when ASDS loss is used, while it remains in the same location when the JVSP loss is employed. We utilize both JVSP loss and ASDS loss for optimization. As shown in the 5th row (labeled as "DiffSketcher"), the combination of these two losses leads to synthesized sketches with more intricate details and a visually more realistic appearance.

## 6  Conclusion

In this work, we have proposed a novel and effective approach to bridge the gap between natural language and free-hand sketches. By leveraging the power of pretrained text-to-image diffusion models, we have developed a method that can generate diverse and high-quality vector sketches based on textual input, without the need for large-scale datasets of sketches or sketch-text pairs. We have also explored different aspects of the model design, such as the stroke initialization strategies, choice of loss functions, and properties of strokes in the differentiable rasterizer, which provide valuable insights for future research. With further research and development, our proposed model, DiffSketcher, has the potential to be a valuable tool in various domains, such as design and education. **Limitation.**    Our approach has two limitations. Firstly, there is a lack of correlation between the

text prompt and sketch abstractness, which may result in unsatisfactory sketches. To address this, we suggest establishing a link between the complexity of the text prompt and the number of strokes used. More details can be found in Supp. I. Secondly, the style of the generated sketches is limited. A possible solution is introducing a style transfer module during the sketch synthesis process.

## 7 Acknowledgement

This work is supported by the National Natural Science Foundation of China (No. 62002012, No. 62006012, and No. 62132001) and CCF-Baidu Open Fund. It is also supported in part by The Hong Kong Jockey Club Charities Trust under Grant 2022-0174, in part by the Hong Kong Research Grant Council under General Research Fund (17203023), in part by the Startup Funding and the Seed Funding for Basic Research for New Staff from The University of Hong Kong, and in part by the funding from UBTECH Robotics.

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

## Supplementary

## Overview

This supplementary material is organized into several sections that provide additional details and analysis related to our work on DiffSketcher. Specifically, it will cover the following topics:

- In section A, we provide the implementation details of DiffSketcher.
- In section B, we present a qualitative comparison of our DiffSketcher with another two text-to-SVG methods, CLIPDraw [7] and VectorFusion [13]. We compare results generated by these methods and analyze the differences in terms of visual quality and semantic consistency.
- In section C, we compare sketches generated by our DiffSketcher with those directly sampled from the LDM (*i.e.*, Stable Diffusion [30]) and analyze the differences in their style.
- In section D, we conducted a perceptual study to assess the authenticity of the synthesized sketches.
- In section E, we compare the results of three different strategies for stroke initialization.
- In section F, we visualize how DiffSketcher gradually sketches an object or a scene.
- In section G, we provide example sketches with different stroke widths.
- In section H, we introduce the details of the evaluation metrics used in our experiments.
- In section I, we show several examples of failure cases.

## A  Implementation Details of DiffSketcher.

We begin by describing the vanilla version of our approach (Section 4.1.1), which involves sampling an image from the latent diffusion model [30] and then automatically sketching it using DiffSketcher. Specifically, given a text prompt, we use a DDIM solver [38] to sample a raster image from the latent diffusion model in 100 steps with classifier-free guidance [12], using a scale of $\omega = 7.5$. To apply the Augmentation SDS loss (Section 4.1.2), we sample a noise level $t$ from the uniform distribution $\mathcal{U}(0.05, 0.95)$, avoiding very high and low values that can cause numerical instabilities. For classifier-free guidance, we set $\omega = 100$, as we found that a higher guidance weight leads to better sample quality. This is larger than the scale used in image sampling methods, which is likely due to the mode-seeking nature of our objective, leading to over-smoothing at small guidance weights.

In DiffSketcher, the number of strokes $n$ is defined by the user, and we use 4 duplicates for image augmentation to maintain recognizability under various distortions. For this purpose, we apply the *torch.transforms.RandomPerspective*, *torch.transforms.RandomResizedCrop*, and *torch.transforms.RandomAdjustSharpness* functions in sequence. It is worth noting that data augmentation is not the focus of our work, and experimental results show that the choice of augmentation strategies does not affect results significantly.

In Section 4, we define a sketch as a set of $n$ strokes $\{s_1, \ldots, s_n\}$ with the opacity attribute placed on a white background. To optimize the control points and opacity, we use two Adam optimizers. Specifically, we set the learning rate of the control point optimizer to 1.0 and the color optimizer to 0.1.

## B  Comparison to Existing Text-to-SVG Work.

This section presents a qualitative comparison between DiffSketcher and CLIPDraw [7]. CLIPDraw is a CLIP-based method introduced for text-to-SVG generation. It gradually optimizes the position and colors of the curves by the gradient descents computed by comparing the cosine similarity of the text prompt and the generated drawings. Our method differs from CLIPDraw in two ways, one is the stroke initialization, and more importantly, ours is equipped with the Augmentation SDS (ASDS) loss. As illustrated in Fig. 9, CLIPDraw struggles to synthesize a meaningful and visually pleasing drawing, no matter whether with colors or not. It can be explained by the fact that the CLIP model is not a generative model, and it can only provide guidance from a highly-semantic perspective. In

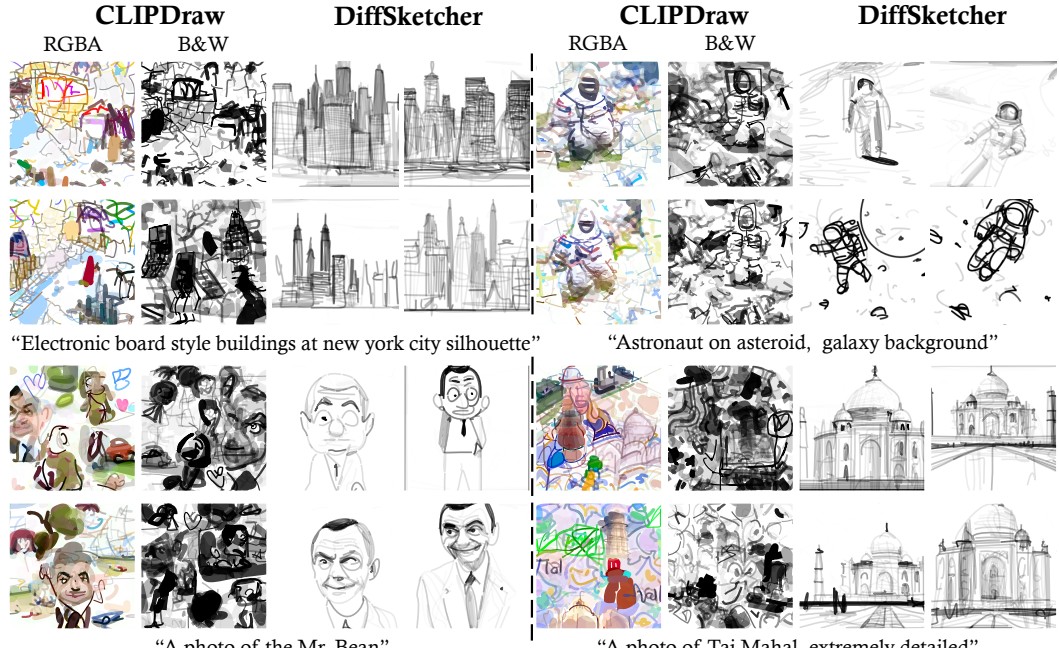

Figure 9: Comparison of the results synthesized by CLIPDraw and DiffSketcher. Specifically, for each example, we compare the results generated by CLIPDraw (Left) and our DiffSketcher (Right) given the same text prompt. We implemented two versions of CLIPDraw: the RGBA version (original version) and the B&W version. The B&W version forces the stroke color to be black to mimic the sketch style. Our results are visually more pleasing and meaningful.

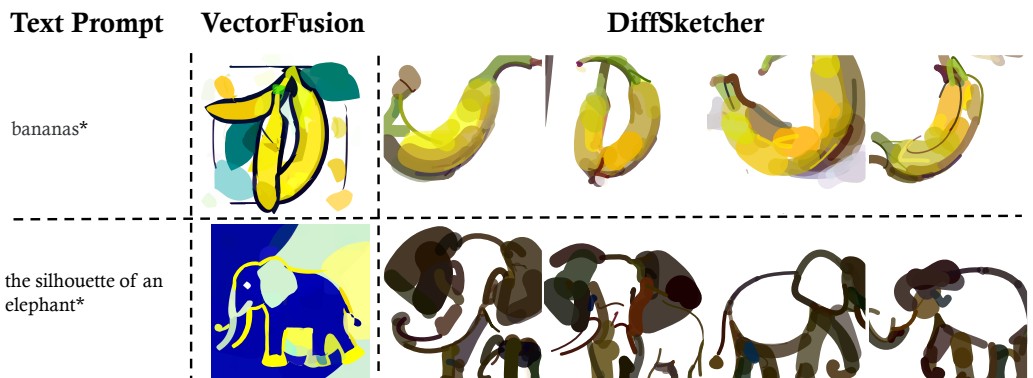

Figure 10: Qualitative comparison with VectorFusion(VF) [13]. VF's results were copied from *Figure 2* of its original paper, with a text prompt suffix of "minimal 2D line drawing trending on ArtStation". In contrast, our results were optimized from scratch using ASDS without specifically designed text prompt suffix.

contrast, our DiffSketcher can generate sketches that are semantically consistent with the input text, and exhibit high aesthetic quality. This is because the proposed ASDS loss can distill the drawing capability of the latent diffusion model (LDM) into the differentiable rasterizer. These results also suggest the effectiveness of the ASDS loss and the benefits of leveraging the power of the LDM. VectorFusion [13] is highly relevant to our work but it aims to produce general vector graphics, such as iconography. Although our proposed method is designed to generate vector sketches, it can easily extend to generate other types of vector graphics. In Fig. 10, we compare results obtained by VectorFusion and our DiffSketcher.

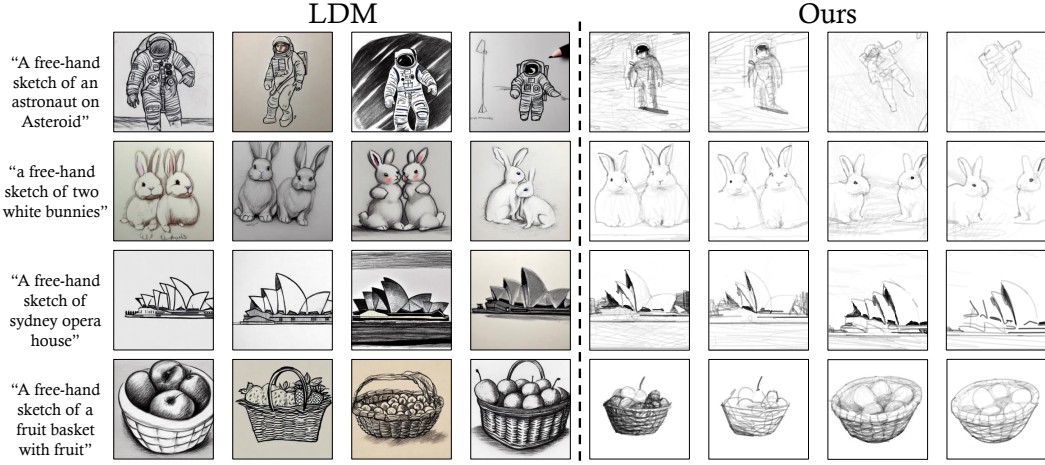

Figure 11: Comparison of sketches generated by sampling from the LDM using the specified text prompt (Left) and ours (Right).

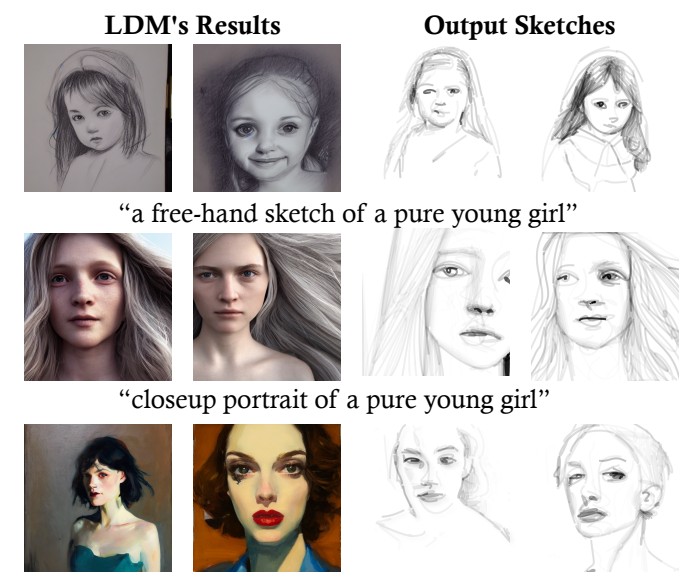

Figure 12: The style of the generated sketches is not significantly affected by the keywords used in the text prompt.

## C  Comparison to Existing Text-to-Image Work.

In this section, we compare the sketches directly generated by LDM using specific text prompts. To encourage the results to be abstract and follow the free-hand sketch style, we append a suffix to the text prompt: "A free-hand sketch of xxx on a white background, trending on ArtStation. Keep abstract." This prompt was tuned qualitatively to capture the desired style and artistic expression.

The results are shown in Fig. 11. It is clear that the LDM is capable of generating high-quality *raster* sketches in a free-hand sketch style. However, different from the *vector* sketches generated by our DiffSketcher, LDM's sketches have two distinguished characteristics: 1. they are more delicate and like professional sketches. 2. their background exhibits the paper texture. We suppose this is because most sketches used for training LDM are photographs of professional sketches drawn on paper. By contrast, the sketches synthesized by our DiffSketcher are more like the style of free-hand sketch and exhibit different levels of abstractness.

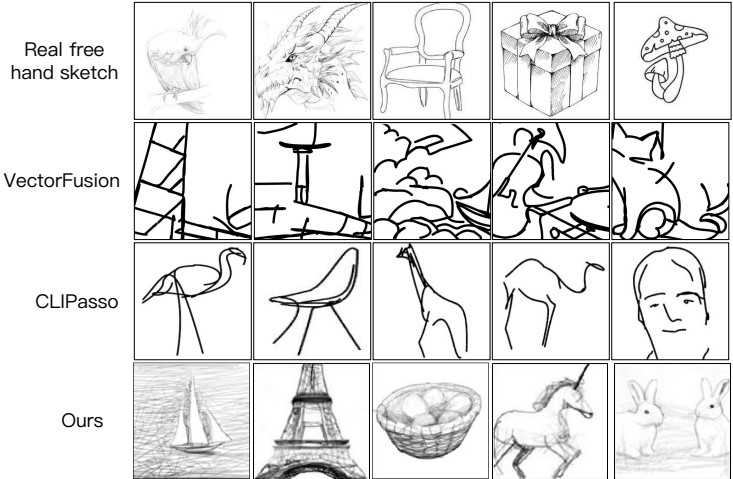

Figure 13: Partial sample visualization for conducting user research. The hand-drawn sketches were sourced from the Google. CLIPasso's and our results were sourced from respective paper, VectorFusion's results were sourced from their project homepage.

It is worth noting that our method does not require indicating "free-hand sketch" in the text prompt. The drawing engine of our model is the rasterizer and it can naturally capture the sketch style. We also conduct experiments to show the results of using different keywords in text prompts, such as "sketch", "drawing", and "photo". The results are shown in Fig. 12. We can see that although the style of the output images is different, the style of the generated sketches is not significantly affected by the keywords.

## D    User Study.

Table 1: Results of the User Study. The Confusion score of real sketch means only 67% real sketches are recognized as real.

| Metric / Method | CLIPasso [45] | VectorFusion [13] | DiffSketcher (Ours) | Human Sketch |
|---|---|---|---|---|
| Confusion Score | 0.39 | 0.33 | 0.65 | 0.67 |

To assess the authenticity of the synthesized sketches, we conducted a perceptual study. Specifically, We gathered a total of 90 synthesized sketches using three different methods (30 samples per method) and obtained 30 real sketches from Google Image by searching for "free-hand sketch". Figure 13 shows a partial sample. We then mixed the real and fake sketches and distributed questionaires to 41 participants. The participants were asked to determine whether each sketch was drawn by a human or not, without any knowledge of its source. We utilized the confusion score as the evaluation metric, where a higher score indicates a greater likelihood of the generated sketches being recognized as real. The results are presented in Table 1. It is clear to see that our method produced sketches that were more frequently identified as real, highlighting the superior quality of our synthesized sketches.

## E    Stroke Initialization.

The highly non-convex nature of the ASDS loss and JVSP loss function makes the optimization process susceptible to stroke initialization, especially for generating scene-level sketches with multi-instances. To address this issue, we explore different stroke initialization strategies (as mentioned in Section 5.3 and Fig. 8) and evaluate their impact on the performance of our model.

As shown in Figure. 14, we compare three different stroke initialization methods: random initialization, initialization based on the CLIP saliency map [45], and initialization based on our proposed fusion attention (ours). The experimental results demonstrate that our proposed strategy, initialization based on fusion attention, outperforms the other methods in terms of visual quality and synthesis efficiency. This initialization method can utilize the joint semantic and structural information from the

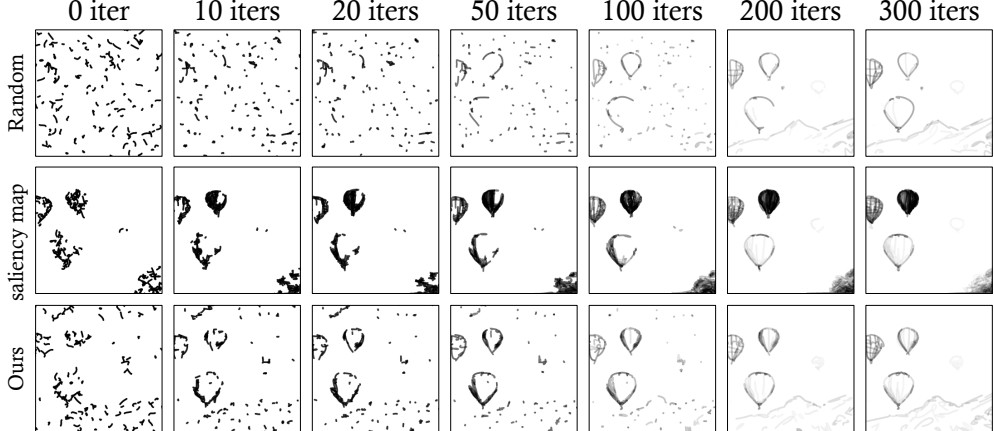

Figure 14: Comparison of the (intermediate) results when using different stroke initialization strategies. From top to bottom: (a) random initialization, (b) initialization based on the CLIP saliency map, (c) our proposed fusion attention.

input text and image (i.e., LDM results) to guide the stroke placement, resulting in more semantically meaningful and artistically expressive sketches. However, using a CLIP saliency map for initialization leads to a sketch with only salient objects while ignoring the background. Random initialization takes more iterations than ours to synthesize visually pleasing results.

## F  Visualization of Sketching Process.

In this section, we show the trace of 300 iters of sketching. By visualizing the intermediate outputs during the generation process, we can gain insights into how our model sketches an object. Specifically, as shown in Fig. 15, we can observe how the strokes are placed and refined over time to gradually form the desired sketch.

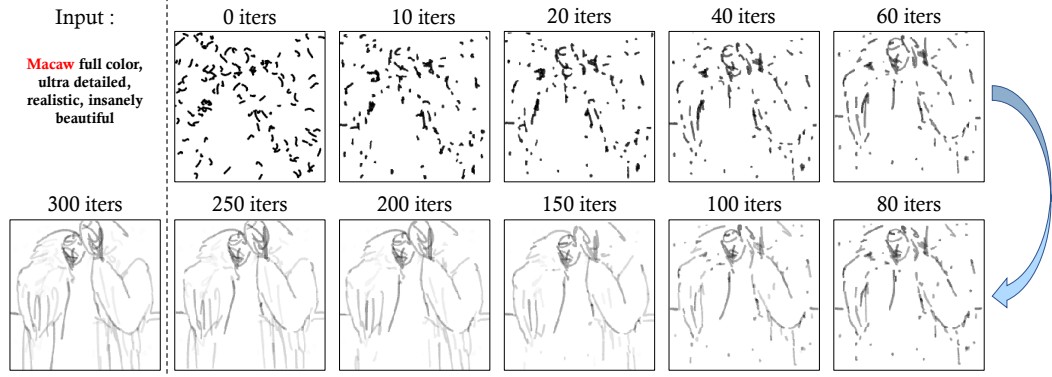

Figure 15: The intermediate results throughout the optimization process.

## G  Effect of the Stroke Width.

In Fig. 16, we compare the results with different stroke widths given the same text prompt. In our implementation, we use a fixed stroke width for all strokes. Such a design is to simplify the optimization, making it computationally more efficient and less prone to overfitting. It can easily extend to include stroke width as a parameter for optimization, like [7, 32].

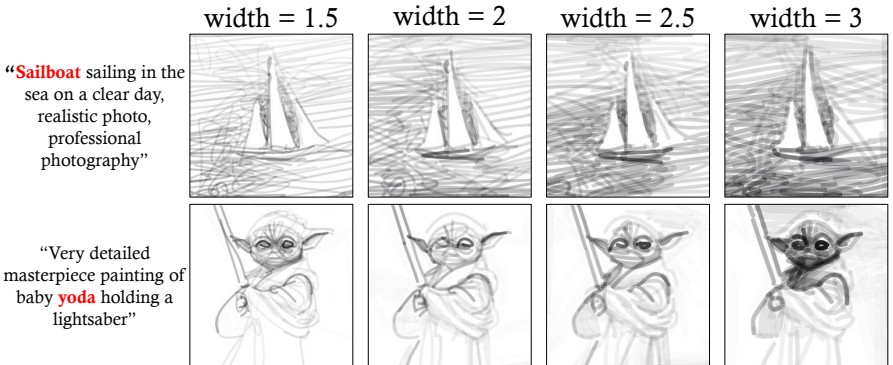

Figure 16: Different widths of the curves. The width increases from left to right.

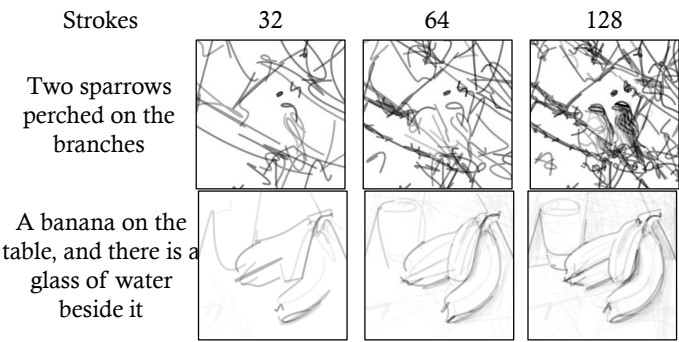

Figure 17: The failure cases.

## H   Evaluation Metrics.

Evaluating text-to-sketch synthesis is challenging due to the absence of ground truth sketches. As we mentioned in Section 5.2 and Fig. 6, we evaluate the models from three aspects: semantic consistency between the generated sketch and text prompt, the aesthetic quality of the sketch, and the recognizability of the sketch.

**Semantic Consistency Between the Generated Sketch and Text Prompt.**   To measure the semantic consistency, namely the CLIP score [30, 7], we calculate the cosine similarity of CLIP ViT-L-14 embeddings of the generated sketches and corresponding input text prompts. Our method achieves a cosine similarity of $0.3494$, which is higher than Canny [2] algorithm ($0.328$) and CLIPasso [45] ($0.3075$).

**The Aesthetic Quality of the Sketch.**   To measure the aesthetic quality of generated sketches, we adopt the CLIP-based aesthetic indicator [33]. This indicator [33] consists of a CLIP ViT-L-14 backbone and a multi-layer perception (MLP), which is pre-trained on LAION [34] data. Figure 5 of the main paper compares the aesthetic score of several examples generated by different methods. Sketches generated by our method obtain the highest scores.

**The Recognizability of the Sketch.**   Finally, to measure the recognizability of the generated sketches, we use the CLIP ViT-L-14 model [27] for zero-shot classification. Specifically, we first generate sketches for 34 categories[2]. Then, we use the CLIP model to classify these sketches. In Fig. 6 of the main paper, we list the probabilities of the sketch being categorized into different classes. Since a Canny edge can preserve the object's contours and fine details, it achieves high recognizability. Compare our DiffSketcher with CLIPasso, our DiffSketcher can draw a more complete sketch. Therefore, our sketches are easier for CLIP to recognize.

---

[2]The 34 categories include "astronaut", "vessel", "observatory", "needle", "outer space", "earth", "iron man", "batman", "apple", "sailboat", "ship", "bunny", "castle", "cabin", "inn", "bike", "cat", "dog", "dragon", "snake", "horse", "fruit basket", "Sydney opera house", "lamp", "lighthouse", "mug", "desk", "macaw", "mountain", "river", "eiffel tower", "unicorn", "yoda", "skyscraper".

# I  Failure Cases

As shown in Fig. 17, our approach mainly has two limitations. Specifically, one limitation is the lack of correlation between the text prompt and sketch abstractness. For instance, if the text prompt describes multiple objects but the number of strokes is set too small, the resulting sketches may be unsatisfactory. A possible solution is to estabilish a link between the complexity of the text prompt (such as the number of described objects) and the number of strokes to be used.

