# OpenReview forum: "DiffSketcher: Text Guided Vector Sketch Synthesis through Latent Diffusion Models"
_NeurIPS.cc/2023/Conference — NeurIPS 2023 poster_

### Official Review · Reviewer_ZP2A · 2023-07-04

**Soundness:** 2 fair
**Presentation:** 2 fair
**Contribution:** 2 fair
**Rating:** 3
**Confidence:** 4

**Summary:**

This paper proposes DiffSketcher, a stroke-based rendering model using a pre-trained text-to-image diffusion model. It performs the task by directly optimizing a set of Bézier curves with an extended version of the score distillation sampling (SDS) loss, which allows using a raster-level diffusion model as a prior for optimizing a parametric vectorized sketch generator.

**Strengths:**

1. This paper proposes a stroke-based rendering method for synthesising vector graphics.
2. This method does not need to retrain the very cost Diffusion model.
3. This method could use text to control the painting content.

**Weaknesses:**

1. Lack of novelty. The model's generation ability essentially origins from the pre-trained diffusion model and the differentiable rasterizer module process is just a fitting process. I will discuss this in the below section "Limitations" further.

2. Lack of comparison. Figure 5 shows the raster images generated by the diffusion model and the corresponding sketching results. However, the compared methods Canny and CLIPasso are too weak. The authors should compare their results with more robust image-to-sketching rendering methods such as "Combining sketch and tone for pencil drawing production" and image style transfer methods such as StyleGan. In the supplementary material, I have seen comparisons with CLIPDraw. But this kind of comparison could not support anything since their artistic styles are totally different.

3. Lack of a very important citation. This paper is a sketching version of the published CVPR 2023 paper "Vectorfusion: Text-to-svg by abstracting pixel-based diffusion models". Both papers follow the same idea that using diffusion models to generate raster images and then using diffvg to optimize stroke parameters. Although the task and technique details are different, no citation of such similar work is unacceptable.

4. Poor writing. The authors should explain their proposed abbreviation the first time they use it. For example, the "ASDS" in line 183. When I got to line 185, I saw the "Augmentation SDS" invented by the author. And I did not see what the abbreviation "JVSP" mean.

**Questions:**

1. What is the total stroke number of a sketch? Is it user-specified or automatically determined by some means?

2. Control points (anchors). In line 225, how do you sample the control points from the distribution map? Using rejection sampling or adaptive sampling or other methods? And the stroke width. In the supplementary material, the authors handly set this parameter. Why not make it learnable as the opacity?

3. Running Efficiency. It seems the optimizing process is very costly. What is the running time to generate a sketch? Please give the used image resolution, stroke number, and iteration steps.

4. In figure 3, is the $\varepsilon$ and $\epsilon$ are two different things?

**Limitations:**

I doubt the meaning of this kind of technique route that uses a pre-trained diffusion model to generate a raster image and then translate the raster image into a vector graphic. This technique route is just an engineering combination of a generative model and a raster-to-vector algorithm. I prefer to see the authors proposing improvements to the diffusion models or the raster-to-vector algorithms, but not just combining them.

---

> ### Author Rebuttal · Authors · 2023-08-10
>
> Thank you for your constructive suggestions. We address your concerns one-by-one below:
>
> **R1-Lack of novelty:** Thanks. Firstly, we respectively disagree with your perspective on the *novelty*. Diffvg] was the first to introduce the differentiable rasterizer, while CLIPasso effectively linked Diffvg and CLIP loss, achieving impressive performance in image-conditioned sketch generation. We believe both works are novel contributions. In this work, we further extend the drawing capabilities of LDM to vector sketches. We introduce to use attention map of LDM to initialize SVG path and present ASDS loss and JVSP loss for optimization. We also provide insights into how each loss affects the synthesis process, and eventually achieve significant progress in the target task. Secondly, although vector graphics are widely used in practice, research on vector graphics generation remains limited. Our work demonstrates a new approach to realize text-based vector sketch generation, which we believe will contribute to the advancement of this research field.
>
> **R2-Lack of comparison:** Thanks for your suggestion. While generative models like GANs and Diffusion Models have shown impressive performance in various domains, it is true that most research, including notable works like StyleGAN, has primarily focused on generating raster images. There has been limited research specifically targeting the generation of vector images such as SVG. CLIPasso is the former SOTA method in this research field, CLIPDraw is also a CLIP-based method specifically designed for text-to-SVG generation, thus we compare with these two methods. Additionally, we follow your suggestion to include VectorFusion for comparison. Please find details in our Common Response.
>
> **R3-Poor writing:** Thank you for pointing it out. JVSP stands for "Joint Visual Semantic and Perceptual loss". This loss is used to optimize the similarity of the synthesized sketches and the instance sampled from the frozen latent diffusion model. ASDS stands for "Augmentated Score Distillation Sampling loss". This loss is utilized to ensure that  the generated images align with the text prompt consistently.
>
> **R4-How to set the stroke number?** In our work, the number of strokes in a sketch is user-specified.
>
> **R5-Parameters to be optimized:** Firstly, we feed the attention map into a softmax layer to obtain the distribution map, and then sample evenly according to their probabilities. Secondly, in our current implementation, the stroke width is preset, meaning it is fixed throughout the sketch generation process. However, it is possible to treat the stroke width as a learnable parameter and optimize it during training. This would allow the model to adaptively adjust the stroke width based on the text input, providing greater control of the generated sketches.
>
> **R6-Running Efficiency:** Follow your suggestion, we tested the rendering efficiency of DiffSketcher on a single NVIDIA A800 and Intel(R) xeon(R) Platinum 8336C CPUs. The results are shown in Table R3:
>
> **Table-R3 Efficiency of our proposed DiffSketcher.**
>
> | Canvas Size | Stroke Number | Iteration Steps | Time (Sec) |
> | --- | --- | --- | --- |
> | 224 * 224 | 32 | 100 | 29 |
> | 224 * 224 | 32 | 200 | 58 |
> | 224 * 224 | 64 | 200 | 60 |
> | 224 * 224 | 64 | 500 | 150 |
> | 224 * 224 | 128 | 200 | 64 |
> | 224 * 224 | 128 | 500 | 160 |
>
> In addition, we conducted a comparison of the rendering time required by VectorFusion to generate a sketch with 64 strokes. VectorFusion uses the same layered rendering strategy as LIVE, and it takes 1404 seconds (with LIVE taking 1149 seconds and LSDS fine-tuning taking 255 seconds) to obtain a reasonable result.
>
> **R7-Typo:** Sorry for the confusion. In figure 3,  $eps$ and $eps$  are the same symbol used to represent Gaussian noise.

---

### Official Review · Reviewer_mvSd · 2023-07-06

**Soundness:** 3 good
**Presentation:** 3 good
**Contribution:** 4 excellent
**Rating:** 7
**Confidence:** 3

**Summary:**

The authors introduce DiffSketcher. The model makes use of a frozen pre-trained latent diffusion model (Imagen) to guide the parametrisation of bezier curves and reproduce the generated image in vectorised sketch form. The method works in a similar way to DreamFusion, but instead of using a differentiable 3D renderer (DreamFusion is a text-to-3D model) the authors use a differentiable bezier curves renderer.

**Strengths:**

- The paper shows very "believable" results: "human-like" sketches.
- The first text-to-sketch model to work at the scene and object level.
- They make good use of the methods available in the literature, the combination of the Li et al [15] differentiable rasteriser with the  score distillation sampling loss from DreamFusion being particularly effective
- The authors additionally improve the random startup of the bezier curves and show how the inclusion of a perceptual (LPIPS) and "semantic" (CLIP feature comparison) loss improves fidelity of the sketch to the prompt.

**Weaknesses:**

- I was not able to identify what is the benefit of using the Augumentation SDS loss instead of the original SDS loss or why it "encourages plausible images to have low loss and implausible ones to have high loss"?
- The authors claim it is possible to control the abstraction level of the sketches but that seems to be through the number of strokes. There are multiple ways of defining abstraction and because these sketches are always based on image inputs it is improbable that the method would be able to reproduce higher abstraction levels such as the ones found on the QuickDraw dataset.
- Some of the notation is confusing. Equations 1, 2 and 3 are probably hard to parse if the reader is not familiar with the original references. The use of S_a and S_phi for sketches and images is also particularly confusing on first read.

**Questions:**

- The current work is constrained by its direct connection to a model that produces realistic images. Do you see how a future method might be able to achieve true abstraction (e.g. cats with whiskers and a house as a triangle+rectangles)?

**Limitations:**

The authors do not discuss current limitations of the model.

---

> ### Author Rebuttal · Authors · 2023-08-10
>
> Thank you for your positive comments and constructive suggestions. We address your concerns one-by-one below:
>
> **R1 - Benefits of ASDS loss:** Thank you for your suggestion. To demonstrate the superiority of ASDS loss over SDS loss, we conducted a new ablation experiment and presented the results in Fig.R5 of the supplementary file. We can observe that using ASDS loss leads to higher visual quality compared to the original SDS loss.
>
> **R2 - Control of the abstraction level of the sketches:** A good question! (1) In this work, we follow CLIPasso to use the number of stroke to control the abstraction level. As mentioned in the Common Response, we acknowledge the need for future research to incorporate automatic control of the stroke count. (2) Regarding reproducing sketches with a higher abstraction level, we show several examples in Fig.R3 first row of the supplementary file. However, our method finds it challening to produce sketches similar to those in the QuickDraw dataset. The sketches from the QuickDraw dataset were drawn by amateurs within a limited time, resulting in significant deformations and abstractions. Consequently, they exhibit a distinct style that differs from ours.
>
> **R3 - Confusing notations:** $S_a$ indicates the augmented version of the sketch, while $S_\phi$ indicates the VAE decoder used to obtain the RGB pixel representation of the sketch.
>
> **R4 - How to achieve true abstraction:** This is a very inspiring question! Firstly, we would like to clarify that when solely using ASDS loss for optimization, our model does not rely on raster images produced by LDM (as explained in Section 4.1.2). However, our model learns how to draw the sketch from the LDM model, which means its drawing ability is limited by the capabilities of the LDM model. Secondly, to achieve the true abstraction, we suppose there are two possible solutions. (1) Treating abstraction, i.e., using basic shapes to represent an object, as a style and introducing a style transfer module to guide the generation model in synthesizing sketches with this specific style. (2) Introducing real-world knowledge to the model, enabling it to understand the composition of each object. This solution is more fundamental. We will explore this problem in our future research.

---

> > ### Comment · Reviewer_mvSd · 2023-08-16
> >
> > Thank you for the thorough response. I have read the other reviews and each of the rebuttals and found that the authors have addressed well the raised concerns; my plan is to keep my current recommendation. It is interesting to know of the existence of the concurrent work of VectorFusion and what are the main differences between the two studies. I am excited to see future work addressing complex abstraction with the authors's suggestions.

---

### Official Review · Reviewer_6RS1 · 2023-07-06

**Soundness:** 3 good
**Presentation:** 3 good
**Contribution:** 3 good
**Rating:** 7
**Confidence:** 4

**Summary:**

The authors introduce DiffSketcher, an algorithm for generating vectorized free-hand sketches from text input. DiffSketcher leverages pretrained text-to-image diffusion models to optimize parametric Bézier curves and generate sketches that align with the text prompt. The method incorporates Score Distillation Sampling (SDS) loss and a differentiable rasterizer to guide the optimization process. Additionally, a joint attention-based stroke initialization strategy improves convergence towards semantic depictions. The paper also demonstrates the effectiveness of DiffSketcher in generating sketches with various levels of abstraction. Overall, this is a promising approach to bridge the gap between natural language and free-hand sketches.

**Strengths:**

- The paper is written really well, and the concepts are explain clearly and intuitively for the most part.
- addressing a relatively underexplored area of text-to-sketch synthesis
- The overall idea is simple (combining differentiable rasterizer with text-to-image diffusion) and it is well-designed specifically for generating vector sketch from text.
- the stroke initialization based on fused attention map is also neat
- interesting idea to incorporate opacity in vector sketch to mimic the way human draw
- the results are good

**Weaknesses:**

- the discussion of the results and comparison is somewhat lacking. For example, the ablation study is really short and I'd love to see more insight on the effect of each components
- The evaluation is also a bit unsatisfying. While the results are impressive on their own, the comparison with existing work seems unfair since the style are quite difference (and if we are using diffusion model to generate image anyway, any image to sketch techniques could have been used?). And I feel like the quantitative evaluation which relies on CLIP embeddings trained on images, may favor sketches with more detailed styles, or more human-like style anyway. So when the baselines are weak, the comparison doesn't really give us much insight into the other strength of the method.
- I also cannot find much information in the paper about the VAE decoder in the proposed method (that convert image to sketch?)

**Questions:**

- Would appreciate some clarification regarding the VAE decoder
- How long does it take per sketch?
- How do you set the number of strokes? Shouldn't complicated prompt need more strokes than simple prompt in order to get the same level of abstraction?

paper that might be related:
Sketchforme: Composing Sketched Scenes from Text Descriptions for Interactive Applications


**Limitations:**

I might have missed it but I don't see any mention of the limitations of the proposed method in the paper. I think this is quite important to see since the evaluation for this sort of task can be quite subjective (qualitative results could have been cherry pick). So I'd appreciate some discussion about the limitation.

---

> ### Author Rebuttal · Authors · 2023-08-10
>
> Thank you for your positive comments and constructive suggestions. We address your concerns one-by-one below:
>
> **R1 - Insufficient discussion about ablation study:** Thank you for your suggestion. From the Fig.6, we can have the following observations: (1) Our proposed initialization strategy significantly accelerates the synthesis process (”CLIPasso+opacity+saliency“ vs. ”CLIPasso+opacity+LDM Attn“); (2) The 4th and 6th rows highlght the effects of JVSP loss and ASDS loss, respectively. When only JVSP loss is used (4th row), the generated sample closely approximates the result of LDM. On the other hand, when only ASDS loss is employed (6th row), the generated sample aligns with the text prompt semantically, but does not follow the attention map of the LDM. (3) Additionally, using ASDS loss results in more diverse sampling outcomes during the synthesis process. For instance, the location of the astronaut's head may change throughout the drawing process when ASDS loss is used, while it remains in the same location when the JVSP loss is employed. (4) Comparing the results of using SDS loss and ASDS loss (newly added), using ASDS loss leads to higher generation quality.
>
> Considering these observations, we can summarize the features of JVSP loss and ASDS loss as follows: (1) JVSP loss: It constrains the alignment between the generated sketch and the LDM's result, making it suitable for a two-stage pipeline. JVSP loss relies on the LDM's result and allows modification through adjustments to the text prompt, providing control over the generated sketch. (2) ASDS loss: It does not rely on the LDM's result and can be used for one-stage optimization. ASDS loss offers more flexibility, resulting in diverse generated samples.
>
> **R2 - Insufficient evaluation:** When submitting this paper, the available baseline methods were very limited. Now we have included a comparison with VectorFusion, whcih is a strong baseline method for comparison. Please find more details in Common Response.
>
> **R3 - Explaination of VAE-decoder:** Sorry for the confusion. In this work, the VAE decoder is not intended for image-to-sketch translation. Instead, we use the open-source stable diffusion model, which is a latent space diffusion model that employs a pre-trained VQ-VAE to encode the image from the pixel space into the latent space to perform the diffusion and denoising process. Consequently, a VQ-VAE is required to decode the latent representation back into the pixel space. We strictly adhere to the decoder parameters specified in stable diffusion model and have updated the Fig.6 in the supplementary file to eliminate any ambiguity.
>
> **R4 - Running Efficiency:** We tested the rendering efficiency of DiffSketcher on a single NVIDIA A800 and Intel(R) xeon(R) Platinum 8336C CPUs. When the number of strokes is 32, it takes 29 seconds to iterate 100 steps and 59 seconds to iterate 200 steps. When the number of strokes is 64, it takes 60 seconds to iterate 200 steps. When the number of strokes is 128, it takes 64 seconds to iterate 200 steps.
>
> **R5 - Control of the number of strokes:**
> Our current method requires the user to mannually set the number of strokes. However, it should be noted that the model can implicitly influence the abstractness of generated sketches by setting the opacity of some strokes to 0 (full transparency) if there are too many strokes. As we discussed in the limitations in the common response, we will leave the automatically control the number of strokes or abstractness of generatd sketches in future research.
>
> **R6 - Missing reference**: Thanks. Sketchforme uses bounding boxes and object sketchers to generate sketch strokes based on text conditions, while our approach directly render vector images based on text prompts. This work is related to our work and we have discussed this paper in related work section of the revised version of the manuscript.

---

> > ### Comment · Reviewer_6RS1 · 2023-08-20
> > **Thanks!**
> >
> > Thank you for the detailed response; it clarifies many of my confusions. While I believe that VectorFusion somewhat diminishes the impact of the work, I still think it remains within the acceptance threshold. Especially given that VectorFusion is a somewhat recent publication. So, although I'm not as optimistic as I was initially, but I'd still like to keep my original rating for now.

---

### Official Review · Reviewer_PspM · 2023-07-06

**Soundness:** 3 good
**Presentation:** 2 fair
**Contribution:** 2 fair
**Rating:** 5
**Confidence:** 3

**Summary:**

This paper proposed an algorithm that creates vectorised free-hand sketches using text as input. Particularly, the authors use a pre-trained text-to-image diffusion model and transfer its prior knowledge into a differentiable rasteriser. It performs the task by directly optimising a set of B\'ezier curves with an extended version of the score distillation sampling (SDS). The loss is backpropagated through the differentiable rasteriser to update the control points and opacity of b\'ezier strokes at each step until convergence of the loss function.

**Strengths:**

The paper is relatively easy to follow and the qualitative results are quite convincing. This line of work -- synthetic sketch generation -- is recently gaining quite some traction and it is nice to see another addition to the community. The authors did a commendable job in writing the paper clearly with nice diagrams that are intuitive.

**Weaknesses:**

The biggest weakness of this paper is its novelty. While there might not be a paper that ONLY focuses on text-to-sketch using SDS loss, the authors missed an important paper that does the same job -- VectorFusion (https://arxiv.org/pdf/2211.11319.pdf).

Particularly, VectorFusion uses the same differentiable rasteriser (including optimising the opacity parameter) to generate 2D line drawings. See Fig. 2 in VectorFusion and section 5.5

Like in this paper, VectorFusion can also control the level of detail using B\'ezier controls. I would love to think that DiffSketcher and VectorFusion are parallel works (and the authors did not know about VectorFusion paper). Nonetheless, it should be noted that VectorFusion was uploaded on arXiv on Nov 21st, 2022 and accepted to CVPR'23 long before NeurIPS submission deadline.

**Questions:**

Apart from my aforementioned weakness, there are a few additional questions/confusions:

First, the authors mention in L3 that the "algorithm that creates vectorized free-hand sketches" -- I think we should carefully consider the definition of what is a "free-hand sketch" -- a sketch that has stroke deformations typically seen in a human drawn sketch.
As DiffSketcher was never trained on a human-drawn sketch, so how can the authors assert that it is generating free-hand sketch and not something that looks realistic but follows edge-maps?
One way of evaluating if DiffSketcher truly generates free-hand sketch and not some edge-map is to train a Fine-Grained Sketch-based Image Retrieval (FG-SBIR) model on sketches from DiffSketcher and evaluate on real human drawn free-hand sketches. The authors can use some standard FG-SBIR dataset like Sketchy or QMUL-Shoe-V2.

On a minor note, the authors should provide a clear description of why SDS is better than something like CLIP. This description could be accompanied with some insightful analysis or experiment. I would leave this to the authors to decide and think what experiments or analysis to show that helps to clarify SDS vs CLIP for DiffSketcher.

Also the authors should better motivate "LDM for Vector Sketch Synthesis". Although I know it is not true but the current introduction reads like -- LDM is a hot topic, hence we extended CLIPasso with LDM.

Finally, to make the paper more interesting, the authors can add an interesting discussion as to why something super simple like "StableDiffusion+CLIPasso" is not a good alternative for text-to-sketch generation.

**Limitations:**

Since this work is quite literally a sub-section of another paper "VectorFusion", it shares the same as mentioned in the discussion section 6 of the paper. I am simply quoting those limitations (application to this paper as well) for completeness:

"VectorFusion faces certain limitations. For instance, forward passes through the generative model are more computationally expensive than contrastive approaches due to its increased capacity. VectorFusion is also inherently limited by Stable Diffusion in terms of dataset biases and quality, though we expect that as text-to-image models advance, VectorFusion will likewise continue to improve"

---

> ### Author Rebuttal · Authors · 2023-08-10
>
> Thank you for your constructive comments. We address your concerns one-by-one below
>
> **R1 - Definition of "Free-hand sketch" and evaluation of these generated sketches:** We appreciate the reviewer for raising this question. However, we respectfully disagree with the reviewer's perspective on the definition of free-hand sketch. Our understanding aligns with the definition provided in CLIPasso that "using a set of thin, black strokes (Bézier curves) placed on a white background". It is important to note that sketches can exhibit different styles depending on the drawing skills of the individuals creating them. Datasets such as Sketch or QMUL-Shoe-V2 contain sketches drawn by amateurs, commonly referred to as amateur sketches. These sketches often exhibit significant deformation and a more abstract nature. In contrast, sketches produced by professionals tend to have fewer deformations and are less abstract. The style of sketches generated by our approach closely resembles that of professional sketches.
>
> To assess the authenticity of the synthesized sketches, we conducted a perceptual study. Specifically, We gathered a total of 90 synthesized sketches using three different methods (30 samples per method) and obtained 30 real sketches from Google Image by searching for "free-hand sketch". We then mixed the real and fake sketches and distributed questionaires to 41 participants. The participants were asked to determine whether each sketch was drawn by a human or not, without any knowledge of its source. We utilized the confusion score as the evaluation metric, where a higher score indicates a greater likelihood of the generated sketches being recognized as real. The results are presented in Table R2. It is clear to see that our method produced sketches that were more frequently identified as real, highlighting the superior quality of our synthesized sketches.
>
> **Table R2 Results of the User Study.**
> | Metric \ Method | CLIPasso |  VectorFusion | DiffSketcher (Ours) | Human Sketch |
> | --- | --- | --- | --- | --- |
> | Confusion Score | 0.39 | 0.33 | 0.65 | 0.67 |
>
> > The Confusion score of real sketch means only 67% real sketches are recognized as real.
>
> **R2 - Comparison of JVSP loss and ASDS loss:** In DiffSketcher,our JVSP loss incorporates both CLIP loss and LPIPS loss, and it is important to note that **ASDS and JVSP do not conflict with each other**. As depicted in Fig. 6 of the manuscript, the 4th and 6th rows highlght the effects of JVSP loss and ASDS loss, respectively. When only JVSP loss is used (4th row), the generated sample closely approximates the result of LDM. On the other hand, when only ASDS loss is employed (6th row), the generated sample aligns with the text prompt semantically, but does not follow the attention map of the LDM. Additionally, using ASDS loss results in more diverse sampling outcomes during the synthesis process. For instance, the location of the astronaut's head may change throughout the drawing process when ASDS loss is used, while it remains in the same location when the JVSP loss is employed.
>
> Considering these observations, we can summarize the features of JVSP loss and ASDS loss as follows: (1) JVSP loss: It constrains the alignment between the generated sketch and the LDM's result, making it suitable for a two-stage pipeline. JVSP loss relies on the LDM's result and allows modification through adjustments to the text prompt, providing control over the generated sketch. (2) ASDS loss: It does not rely on the LDM's result and can be used for one-stage optimization. ASDS loss offers more flexibility, resulting in diverse generated samples.
>
> In our full model, we leverage the advantages of both JVSP loss and ASDS loss by employing them for optimization. As demonstrated in the 5th row of our results (labeled "DiffSketcher"), when these two losses are combined, the synthesized sketches exhibit more intricate details and appear visually more realistic.
>
> **R3 - Motivation of our approach:** Thanks. The motivation behind that we develop our model based on LDM lies in three aspects: (1) LDM shows great potential in raster image generation. Unfortunately, it cannot generate vector sketches. (2) CLIPasso achieves great performance in image-conditioned sketch generation, inspiring us to utilize differentiable rasterizer for vector sketch generation. (3) DreamFusion shows a way to distill the drawing knowledge from the LDM to Nerf, which inspires us to transfer the knowledge from the LDM to the differentiable rasterizer. We will modify Line 38~39 to provide a clearer motivation.
>
> **R4 - Why "StableDiffusion+CLIPasso" is not a good alternative:** A good question! There are primarily two reasons: Firstly, the original CLIPasso does not incorporate opacity for optimization, resulting in relatively low visual quality of the synthesized sketches, as depicted in Fig. 5 of the manuscript. Secondly, the optimization process of CLIPasso is time-consuming. As illustrated in Fig. 6 (labled "CLIPasso+opacity+saliency"), approximately 2000 steps are required to achieve a satisfactory result. It is worth noting that the inclusion of opacity in the optimization process enhances the visual quality of the result. For comparison, we replace with our proposed initialization strategy (labeled "CLIPasso+opacity+LDM Attn"), the results show that the synthesis process can be accelated by approximately 8 times.

---

> > ### Comment · Reviewer_PspM · 2023-08-16
> >
> > I would like to thank the authors for a detailed response to all my queries, for which I am leaning towards increasing my initial rating. Keeping minor queries aside, I am mostly focused on the significance of DiffSketcher in the context of VectorFusion. I understand that the authors were not aware of VectorFusion. Nonetheless, given that VectorFusion was available for over 5 months, I would like to clarify a few things.
> >
> > The entire argument boils down to two key factors where the proposed method is different from existing CLIPasso and VectorFusion: (i) adding opacity in the optimisation process, (ii) a faster optimisation using LDM Attn instead of CLIP-saliency or LIVE.
> >
> > ### For opacity:
> > The authors are interested in -- "sketches produced by professionals tend to have fewer deformations and are less abstract. The style of sketches generated by our approach closely resembles that of professional sketches".
> >
> > Adding opacity leads to a lot of noisy strokes compared to the cleaner-looking sketch generated by VectorFusion -- hence, is VectorFusion more "professional looking" than DiffSketcher? This question is also relevant for the human study where the subjects were asked: "whether each sketch was drawn by a human or not". Would this study lead to a different conclusion if the question was reframed to "whether each sketch was drawn by a **professional** or not"?
> >
> > ### For faster optimisation
> >
> > While it is true that DiffSketcher is faster than VectorFusion, we must note that VectorFusion is a more generic approach (encompassing all vector graphics where DiffSketcher gives poor results). This begs the question -- is DiffSketcher a simplified version of VectorFusion (albeit with SDF Attn instead of LIVE for initialisation) which allows faster optimisation?

---

> > > ### Author Response · Authors · 2023-08-17
> > > **Responses to Reviewer PspM's Comments.**
> > >
> > > We would like to express our gratitude for your detailed feedback and for re-evaluating your rating of our work. Regarding the questions mentioned in your response, we would like to clarify three things:
> > >
> > > 1. **Differences between our work and VectorFusion:** As mentioned in the Common Response, besides optimization parameters and initialization strategy, our model also differs from VectorFusion in terms of prompt design and optimization strategy. It is worth noting that our model can synthesize different types of vector graphics using the same pipeline, while VectorFusion requires the use of different model variants to generate different types of vector graphics due to the limitation of LIVE. Additionally, our model utilizes both ASDS loss and JVSP loss for optimization, and we provide insights into how each loss affects the drawing results. We believe that these technical designs and discussions will benefit the development of the community. **Differences between our work and CLIPasso:** While both our method and CLIPasso utilize the differentiable rasterizer for image rasterization, our work and CLIPasso have distinct goals, and thus the pipelines are significantly different.
> > > 2. **Clarification of the results of DiffSketcher:** We respectfully disagree with the reviewer's comments on our results, stating that "DiffSketcher gives poor results." Although our method was originally designed for generating sketch-style vector images, it is indeed capable of generalizing to synthesize other styles of images, such as icon-style and oil-painting-style. As demonstrated in Fig.4 of the supplementary material accompanying our response, these images exhibit high visual quality. It is worth noting that when generating different types of vector graphics, our model utilizes the same pipeline with the same primitive (i.e., Bezier curve).
> > > 3. **Which more closely resembles a professional freehand sketch: DiffSketcher or VectorFusion?** By including opacity for optimization, our model can simulate different styles of strokes that result from various levels of pressure, which are especially common in professional sketches. This enables our results to closely resemble a professional freehand sketch. Additionally, following your suggestion, we conducted another user study in which we replaced the question in our original study with the new question. This time, we collected real sketches by searching for "professional freehand sketch" and compared them with the fake sketches generated by DiffSketcher and VectorFusion. We will update the results once we have obtained them.
> > >
> > > Once again, we appreciate your insightful comments and will address these concerns in the revised version of our manuscript. If you have any additional concerns or suggestions, please do not hesitate to let us know.

---

> > > ### Author Response · Authors · 2023-08-19
> > > **Added User Study**
> > >
> > > **Table R3. The results of the user study.**
> > >
> > > | Metric \ Method |  VectorFusion | DiffSketcher | Human Sketch |
> > > | --- | --- | --- | --- |
> > > | Confusion Score. | 0.124 | 0.625 | 0.689 |
> > >
> > > > The confusion score for real sketches indicates that 68.9% of real sketches are recognized as drawings by professionals.
> > >
> > > These results are consistent with the previous user study and suggest that our results more closely resemble a real professional freehand sketch than those produced by VectorFusion.

---

> > > > ### Comment · Reviewer_PspM · 2023-08-20
> > > >
> > > > I would like to thank the authors for their response and the additional human study.
> > > >
> > > > It would be good to include (i) both of these human studies and (ii) a detailed comparison and differences to VectorFusion in the supplemental. The authors can cite VectorFusion and mention that DiffSketcher is a parallel work.

---

### Official Review · Reviewer_iPV2 · 2023-07-07

**Soundness:** 4 excellent
**Presentation:** 4 excellent
**Contribution:** 3 good
**Rating:** 7
**Confidence:** 4

**Summary:**

This paper proposes an algorithm to leverage pretrained image diffusion models for text-guided sketch synthesis. The sketch to be optimized is represented as a set of Bezier curves (and their opacities) and optimized with the (augmentation) score-distillation sampling (ASDS) loss. Attention maps from the diffusion model's UNet are fused and used to initialize the Bezier curves to accelerate convergence. A variety of results ranging from abstract to concrete sketch generation are demonstrated.

**Strengths:**

The paper is easy to understand and makes a first-of-its-kind contribution: using pretrained image diffusion models to optimize vector graphics sketches. While it is thematically similar to DreamFusion and related works, the problem and solution are novel enough and interesting. The results are convincing are significantly better than prior work, and evaluation is comprehensive on a wide range of concepts. The ASDS loss is interesting although a similar augmentation strategy was also used in CLIPDraw. The stroke initialization strategy by analyzing the attention maps is an excellent idea and could potentially be helpful in other future works that have to deal with domain gaps and difficult optimization landscapes. Ablation studies are quite comprehensive. Overall, the paper is technically strong and demonstrates excellent results.

**Weaknesses:**

This is a strong paper, I do not see any major weaknesses. It would be good to report some failure cases and discuss limitations.


**Questions:**

Minor suggestions:
Line 177: this is good place to define the JVSP abbreviation
Line 183: the abbreviation ASDS is not defined


**Limitations:**

Limitations are not discussed, it would be good to include those in the revision along with some failure cases.

---

> ### Author Rebuttal · Authors · 2023-08-10
>
> Thank you for your positive comments and suggestions. Please see our discussion about limitations in the Common Response.
>
> **R1 - Definition of JVSP and ASDS:** Thank you for pointing it out. In our work, JVSP stands for "Joint Visual Semantic and Perceptual loss". This loss is used to optimize the similarity of the synthesized sketches and the instance sampled from the frozen latent diffusion model. ASDS stands for "Augmentated Score Distillation Sampling loss". This loss is utilized to ensure that  the generated images align with the text prompt consistently. We have included these denifitions in Line179-189 of Sec.4 in the revised version of the manuscript.

---

> > ### Comment · Reviewer_iPV2 · 2023-08-20
> >
> > Thanks for the response. The limitations are all sensible and it would be good to add these to the paper. I will retain my score and recommend acceptance.

---

### Author Rebuttal · Authors · 2023-08-10

We appreciate the reviewers for their constructive comments. Firstly, we would like to address the common concern raised by Reviewer PspM and Reviewer ZP2A regarding the missing reference of the work VectorFusion, and then discuss the limitations of our approach.

**Comparison with VectorFusion (CVPR'23):** Indeed, we were not aware of this paper prior to our submission, and we only found it when it was listed in   the program on the CVPR 2023 website.

VectorFusion is highly relevant to our work. However, there are notable differences between our appraoch and VectorFusion in terms of task setting, methodology, and performance. Firstly, our method, DiffSketcher, primarily focuses on generating vector sketches based on text input. In contrast, VectorFusion aims to generate a broader range of vector graphics. It is important to highlight that our method has the capability to easily extend its functionality to generate other types of vector graphics, such as icon-style image, without the need to alter primitives, we show the results of our approach in generating general vector graphics in Fig.4 of the supplementary file.

Secondly, DiffSketcher follows a distinct pipeline compared to VectorFusion. VectorFusion employs different model variants for different types of vector graphics. For generating vector sketches, our approach differs from VectorFusion mainly in two aspects: **1) Initialization:** VectorFusion randomly initializes the rasterizer (i.e., the location of the control points), while our approach utilizes the attention layer of LDM, guided by text prompts, for initialization. This significantly improve both efficiency and syntehsis quality. **2) Optimization:** While VectorFusion solely optimizes the position of control points, our method also optimize the opacity of stroke. This contributes to enhance the visual quality of the synthesized sketches. We present a performance comparison between these two methods in terms of vector sketch generation in Fig.1.

Furthremore, when generating other types of vector graphics, VectorFusion involves generating raster images using LDM and subsequently converting them into vector images using LIVE. However, our approach does not require the conversion step. Besides, VectorFusion necessitates chaning the primitives and specifically designed prompt suffix string for generating different types of vector graphics, whereas our approach maintains a consistent primitive (i.e., Bézier curve) and has no limitations on text prompt. In Table R1, we provide a comprehensive comparison of these two methods, highlightling their differences and distinctive features.

**Table-R1 Comparison of our approach and VectorFusion.**

|  | DiffSketcher (ours) | VectorFusion |
| --- | --- | --- |
| Prompt | Any prompt, no limitations | Prompt with specifically designed suffix string |
| Initialization | Fusing cross-attention and self-attention of LDM to initialize control points | Using LIVE to convert the raster image to vector format for initlization |
| Optimization parameters | Control points + Opacity| Control points |
| Optimization | All SVG paths are optimized together | Follow LIVE, using layer-wise optimization (8 paths at a time) |
| Efficiency | 150 seconds per SVG | 1400 seconds per SVG |

**Limitations of the proposed method:** Our approach mainly has two limitations. Specifically, one limitation is the lack of correlation between the text prompt and sketch abstractness. For instance, if the text prompt describes multiple objects but the number of strokes is set too small, the resulting sketches may be unsatisfactory. A possible solution is to estabilish a link between the complexity of the text prompt (such as the number of described objects) and the number of strokes to be used. We show several examples of failure cases (Fig.3) in supplementary file.

Another limitation of the current approach is the style of the generated sketches is limited. We can address this by introducing a style transfer module during the sketch synthesis process.

---

### Author Response · Authors · 2023-08-19
**Any unclear explanations?**

Dear Reviewers,

Thanks for your efforts in reviewing this paper. We have tried our best to address the concerns and improve our work. Are there unclear explanations here? We can further clarify them.

Best wishes,

Authors

---

### Decision · Program_Chairs · 2023-09-21

**Decision:**

Accept (poster)

**Comment:**

The paper proposes a method for generating sketch-like images from text by using Score Distillation Sampling (SDS) on a pretrained text-to-image model to directly optimize a set of Bezier curves. Reviewers were generally favorable on the paper, and appreciated the method and results.

The primary concern raised by reviewers was a missing citation and comparison to VectorFusion which was published at CVPR 2023. In their rebuttal, the authors provided a detailed comparison of their approach with VectorFusion, including both a description of the technical differences between the methods and a user study demonstrating strong performance of the proposed method. Reviewers were largely sympathetic to these arguments, and in the end all reviewers were in favor of acceptance.

The AC agrees that the proposed method is distinct enough from VectorFusion to be of interest, and finds the additional discussion and comparisons provided in the rebuttal to be convincing. The authors are strongly encouraged to include these comparisons and results in the camera-ready version of the paper.